# The Role of Apolipoproteins in the Commonest Cancers: A Review

**DOI:** 10.3390/cancers15235565

**Published:** 2023-11-24

**Authors:** Nour M. Darwish, Mooza Kh. Al-Hail, Youssef Mohamed, Rafif Al Saady, Sara Mohsen, Amna Zar, Layla Al-Mansoori, Shona Pedersen

**Affiliations:** 1College of Medicine, QU Health, Qatar University, Doha P.O. Box 2713, Qatar; nd2007863@student.qu.edu.qa (N.M.D.); ma1901280@student.qu.edu.qa (M.K.A.-H.); ye2205159@student.qu.edu.qa (Y.M.); rafif@qu.edu.qa (R.A.S.); sm2104138@student.qu.edu.qa (S.M.); ar2104204@student.qu.edu.qa (A.Z.); 2Biomedical Research Center, Qatar University, Doha P.O. Box 2713, Qatar; almansouri@qu.edu.qa

**Keywords:** apolipoproteins, carcinogenesis, gene expression, therapeutics

## Abstract

**Simple Summary:**

Apolipoproteins (APOs) are crucial components in our blood that are responsible for fat management. Recent scientific discoveries suggest a provocative link between APOs and a range of cancers, including, but not limited to, breast, lung, and prostate cancer. The specific role of some APOs in causing cancer remains enigmatic. In this review, we summarize evidence from the literature supporting the potential involvement of APOs in the onset of cancer. We also highlight promising avenues for treatment through the inhibition of APOs.

**Abstract:**

Apolipoproteins (APOs) are vital structural components of plasma lipoproteins that are involved in lipid metabolism and transport. Recent studies have reported an association between apolipoprotein dysregulation and the onset of a variety of human cancers; however, the role of certain APOs in cancer development remains unknown. Based on recent work, we hypothesize that APOs might be involved in the onset of cancer, with a focus on the most common cancers, including breast, lung, gynecological, colorectal, thyroid, gastric, pancreatic, hepatic, and prostate cancers. This review will focus on the evidence supporting this hypothesis, the mechanisms linking APOs to the onset of cancer, and the potential clinical relevance of its various inhibitors.

## 1. Introduction

Apolipoproteins (APOs) are proteins that bind to lipids to form lipoproteins, and are mainly synthesized in the liver and intestine [1]. APOs are conventionally classified as either insoluble and soluble [2]. The lipid component of lipoproteins is insoluble in water; insoluble APOs are constantly attached to the lipoprotein molecule and cannot endure in the plasma [2]. However, due to their amphipathic nature, apolipoproteins and other amphipathic molecules can surround lipids to synthesize a water-soluble lipoprotein that can be carried through blood or lymph [2]. Figure 1 illustrates the general structure of apolipoproteins. APOs act as ligands for cell membrane receptors, enzyme cofactors, and lipoproteins’ structural components by acting as lipid carriers [3,4]. Based on the densities of the formed lipoproteins, they are divided into five types, including chylomicrons (CMs), very-low-density lipoprotein (VLDL), low-density lipoprotein (LDL), intermediate-density lipoprotein (IDL), and high-density lipoprotein (HDL) [5,6].

The gene family of APOs in humans consists of 22 genes, including APOA1, APOA2, APOA4, APOA5, APOB-48, APOB-100, APOC1, APOC2, APOC3, APOC4, APOD, APOE, APOH, APOL1, APOL2, APOL3, APOL4, APOL5, APOL6, APOM, APOO, and APOJ, which are further classified into 10 subfamilies (APOA-APOJ) [7,8]. APO functions can be altered by gene mutations during their synthesis, leading to the manifestation of different diseases such as hyperlipidemia, atherosclerosis, and coronary artery disease [9,10]. In addition, APOs are associated with the onset and progression of several cancers, due to either their overexpression or loss of expression [7].

As shown in Table 1, the level of APOs in different cancers has been studied to identify them as possible biomarkers. In this review, we recapitulate the current literature on the cellular mechanisms involving APOs in different cancers. We will also discuss the many roles played by APOs in the onset and progression of cancer and their potential to act as possible cancer biomarkers. 

## 2. Functions of Apolipoproteins

APOs are physiologically expressed in both normal and malignant cells; consequently, the following sections will discuss the function of APOs in both contexts.

## 3. Functions of Apolipoproteins in Normal Cells

APOs play a vital role in normal human vascular biology, lipoprotein metabolism, and lipid transport [11]. Plasma APOs bind to the lipoprotein surface and stabilize its structure, and also act as a cofactor for enzymes, regulate enzyme activity, and induce lipid metabolism [3,4,11]. APOs’ configuration and content influence lipoprotein formation and metabolism. APOs bind to membrane lipoprotein receptors and regulate the cellular uptake of lipoproteins. For example, while APOB-100 and APOE bind to the LDL (APOB/E) receptor, APOE binds to the LDL receptor-related protein (LRP) of the liver and extrahepatic tissues, where they are responsible for the redistribution of cholesterol among cells for use in membrane biosynthesis and as a precursor for steroid production [12,13]. On the other hand, APOC2 binds to triglycerides (TGLs), chylomicrons (CMs), very-low-density lipoproteins (VLDLs), low-density lipoproteins (LDLs), and high-density lipoproteins (HDLs) in plasma [14]. 

The three main stages of lipid metabolism include the distribution of triglycerides between organs (fuel transport pathway), the maintenance of the extracellular cholesterol pool (overflow pathway), and HDL metabolism (cholesterol transport reversal) [11]. 

Lipoproteins can be metabolized via two pathways: an exogenous and an endogenous pathway (Figure 2). The exogenous pathway delivers triglycerides to the peripheral tissues via chylomicrons and VLDLs [8]. The enterocytes reintegrate triacylglycerols, phospholipids, and cholesterol on APOB-48 into chylomicron particles; these particles are released into the lymph and reach the plasma via the thoracic duct [8]. Precursor chylomicrons primarily consist of APOB-48 in addition to APOA1, APOA2, APOA4, APOA5, APOC1, APOC2, APOC3, and APOE, which form the surface of the chylomicrons [15]. At peripheral tissues, lipoprotein lipase (LPL) on blood vessel walls breaks down chylomicron triacylglycerols into fatty acids and glycerol for tissue absorption and use or storage. Post lipid release, chylomicrons shrink, becoming remnants that the liver absorbs via LDL receptors and LRP. APOE and APOB48 on remnants aid liver uptake, while APOA and APOC return to HDL in the blood. LDL’s role is to transport cholesterol to cells, not to metabolize chylomicron fats [16]. 

In contrast, in the endogenous pathway, hepatocytes produce triglycerides de novo, which are transported to the peripheral tissues by the VLDL particles [15]. APOB100 is the only component of precursor VLDLs. APOB lipidation is induced by the microsomal triglyceride transfer protein. Once VLDL enters the plasma, it acquires APOs (APOA1, APOA2, APOA4, APOC1, APOC2, APOC3, and APOE) as well as cholesteryl esters from HDL [15]. Like chylomicrons, in the peripheral tissues, VLDL triglycerols are partially hydrolyzed by lipoprotein lipase (LPL) to produce VLDL remnants that only bind to APOE and are thus absorbed by the liver or further digested by hepatic triglyceride lipase, transforming them into LDL [17,18,19]. 

Lastly, reverse cholesterol transport and cholesterol recycling are responsible for the metabolism of HDL and are involved in the removal of cholesterol from peripheral cells to the liver [20,21,22]. The compiled HDL on APOA is covalently modified and converted into precursor HDL, and released as phospholipid-rich, disc-shaped particles by the liver and intestine [15]. Nascent HDL particles absorb free cholesterol from cells by binding to ATP-binding cassette transporter A1 (ABCA1) along with APOA1 and APOA4 [15]. The major APO component of HDL, APOA1, triggers lecithin cholesterol acyltransferase (LCAT), which induces the esterification of free cholesterol and transforms HDL3 particles to larger particles (HDL2) through the accumulation of apolipoproteins (APOA, APOC, and APOE), cholesteryl ester, and triglycerides [23]. The reverse cholesterol transport can occur via three different pathways. First, HDL2 particles with multiple APOE copies are absorbed into the liver by the LDL receptor [20]. The second pathway involves the scavenger receptor B1-mediated absorption of cholesteryl esters from HDL by the liver [24]. Lastly, the third plausible pathway is the transfer of the cholesteryl esters from HDL to triglyceride-rich lipoproteins by the cholesteryl ester transfer protein [20]. 

Additionally, macrophages express APOC2, and since macrophages consume large amounts of energy, APOC2 aids in the transport of lipids into macrophage cells. Enhanced STAT1 protein synthesis induces the overexpression of APOC2, as shown in studies on mice [25]. 

The disruption of APO concentrations, regulating metabolism, enzyme functions, or lipoprotein production, can significantly affect the antiatherogenic properties of HDL and lead to the onset of cardiovascular disease, diabetes mellitus, and obesity [26,27].

## 4. Role of Apolipoproteins in Cancer

Lipid metabolism is considered one of the significantly affected metabolic pathways in cancer [28]. Additionally, several studies have reported a correlation between lipids and APOs in cancer onset and development. The following sections will discuss the presence and roles of different APOs in the commonest human cancers.

## 5. Role of Apolipoproteins in Breast Cancer

The APOA subtype APOA1 has a tumor-suppressive role in breast cancer and plays a role in inducing apoptosis, thus inhibiting the progression of cancer cells [29]. APOA is an essential apolipoprotein that plays a significant role in reverse cholesterol transport [30]. In cancer cells, the interaction between complement component 1q subcomponent binding protein (C1QBP) and APOA leads to the binding of C1QBP to APOA, inhibiting its expression and weakening APOA’s antioxidation ability, leading to carcinogenesis [31]. Additionally, another study has established that APOA1 functions as a cofactor for lectin cholesterol acyltransferase (LCAT), which is a key participant in lipid metabolism. This finding is significant, as lipid metabolism has been extensively linked to cancer in numerous earlier studies. Specifically, the relationship between lipoproteins or lipids and cancer risk has been investigated, and correlations have been established [30]. Studies have shown reduced APOA1 expression in the serum of breast cancer patients, and APOA1 gene mutations are linked with an increased risk of developing breast cancer [32,33,34]. Thus, the normal expression of APOA1 causes cancer cell apoptosis (Figure 3). Nouri et al. [35] conducted a meta-analysis and reported that APOA1 was linked to an increased risk of intraocular metastasis from breast cancer. Contrary to the role of APOA1 as a tumor suppressor in breast cancer, an in vitro study by Cedo et al. [36] demonstrated that APOA1-containing HDL triggered breast tumor development in PyMT mice plausibly due to low oxLDL and 27-hydroxycholesterol levels. 

Mutations in the APOB gene, including 7673CT as well as 12,669 GA, are significantly associated with an increased risk of breast cancer and are frequently present in menopausal females [34]. Furthermore, enhanced APOB expression is considered a statistically significant risk factor for the intraocular metastasis of breast cancer [34]. However, in this study, APOA1 proved to be a more effective biomarker for distinguishing intra-ocular metastasis compared to APOB. According to a recent study, the total cholesterol to APOB ratio did not distinguish between breast cancer patients and control patients [37]. However, the level of APOB was higher in triple-negative breast cancer patients in comparison to other molecular types [37]. Yet their APOB levels were comparable in all stages of breast cancer [37].

Another study revealed that an increase in the APOB to APOA1 ratio is associated with an increase in the severity of breast cancer, but this was not statistically significant [38]. 

In another study, plasma levels of APOC1 were reduced in breast cancer patients [39]. The study further revealed that administering an APOC1 peptide to mice with xenografts exhibited an anti-tumor effect, underscoring the significance of APOC1 in breast cancer development [39]. Despite this crucial discovery, which is echoed in human cases, the specific role of APOC1 in human breast cancer development is still not well-defined. APOC1 testing could also differentiate between triple-negative breast cancer patients and non-triple-negative breast cancer patients [40]. In addition, it has been proven that breast cancer patients have decreased APOC1 and APOC2 but increased APOC3 levels [41]. 

APOD is considered carcinogenic due to its presence in the breast primary cyst fluid content, as these cysts increase the risk of developing breast cancer by threefold [42]. Serum levels of APOD peak when the tumor is benign but reduces with more invasive and metastatic forms of breast tumors [43]. Changes in APOD concentration are influenced by breast cancer metastasis and invasion, but older breast cancer patients also have increased levels compared to younger patients [44]. A recent study concluded that the APOD gene was significantly reduced in breast cancer patients [45] and displays an anti-tumor effect by suppressing MAPK, leading to a restriction in cell mitosis [46]. 

Even though the function of APOE in cancer progression is unclear, it can inhibit proliferation due to its high-affinity interaction with proteoglycan and heparin in the cancer tissue [47]. The association between breast cancer and APOE was controversial in the beginning due to a lack of associations between them; however, research has revealed that patients with one or two copies of the e4 allele along with elevated triglyceride levels were at a fourfold risk of developing breast cancer in comparison to those with low triglyceride concentrations [48,49]. Additionally, researchers have documented contradicting reports between APOE polymorphisms and breast cancer risk; while some studies have reported an association [48,50,51,52,53,54], other studies did not report any link between APOE polymorphism and breast cancer risk [55,56]. A meta-analysis concluded that, among Asians, carriers of the E4E4, E4E3, and E4E2 genotypes were associated with a high breast cancer risk, compared to the E3E3 genotype [57]. Likewise, in Taiwan, studies reported a risk of breast cancer in females with the APOE genotype; neither the APOE2 nor APOE4 alleles showed a notable correlation with markers of cell growth [51,52,58]. On the other hand, Caucasians had no association between breast cancer and APOE2, APOE3, or APOE4 [57]. Moreover, has been APOE positively correlated with breast cancer progression and invasion [47]. 

APOH, a multifunctional apolipoprotein, has been detected in the sera of breast cancer patients [59,60]. Although it was previously established that APOH is elevated in breast cancer, Chung et al. identified that a novel APOH fragment, 3808 Da, was found to be elevated in breast cancer sera [59]. Elevated levels of APOJ were also found in breast cancer patients [61,62]; Yom et al. (2009) [62] reported the overexpression of APOJ in early-stage invasive breast cancer, indicating a role of APOJ in the initiation of breast cancer tumorigenesis. Moreover, the study also documented a high expression of APOJ in patients with <T2 stage breast cancer, thus suggesting the immunostaining of APOJ as a predictive tool in addition to its role as a prognostic factor for recurrence [62]. In vitro studies have shown APOJ knockdown to enhance sensitivity to chemotherapeutic drugs, as well as reduce cell proliferation and metastasis [63,64,65]. 

Although APOLs are involved in various cancers, their role in breast cancer is not well documented [7,66,67,68,69]. Moreover, APOL3 has been found to regulate neuronal calcium sensor 1 (NCS-1), which plays a critical role in promoting the metastasis and survival of breast cancer cells in vitro [70,71]. 

Although APOM is not yet confirmed to be increased or decreased in breast, cervical, and ovarian cancers, some mechanisms explain the plausible inhibition of the tumor growth through APOM [47]. APOM carries and stabilizes the level of sphingosine-1-phosphate, a molecule responsible for reducing cancer invasion [72], thus suggesting the underlying mechanism underpinning APOM-reduced breast cancer growth. A recent study illustrated that APOM expression is statistically significantly lower in breast cancer tissue as compared to normal tissue [73]. In vitro data also showed APOM to suppress breast cancer cell proliferation, migration, and invasion [73]. Table 2 provides a concise overview of the functions of different apolipoproteins in relation to breast cancer. 

## 6. Apolipoproteins in Gynecological Cancers

Studies have reported a role of APOA1 in gynecological cancers. The loss of APOA1, APOA2, and APOA4 was reported in ovarian cancer patients [74]. APOA1 was found to have a suppressive effect on ovarian cancer [74]. In vivo experiments demonstrated the induction of APOA1 in mice to suppress palpable tumors and metastasis, further improving the survival rate by modulating the immune system and altering the host environment. APOA1 changes the phenotypic expression of the macrophages from pro-tumor (M2) to anti-tumor (M1), and blocks tumor-associated angiogenesis by downregulating MMP-9 expression [75]. In ovarian cancer patients, low serum APOA1 levels were detected, suggesting early-stage ovarian cancer, with a sensitivity of 54% and a specificity of 98% [76]. The serum level of APOA4 was noted to be reduced in the serum of patients with ovarian cancer [74].

Additionally, a significant association was reported between increased APOB and high-grade ovarian cancer [77]. This study involved samples from newly diagnosed high-grade ovarian cancer cases and suggested that increased APOB levels might be indicative of a favorable prognosis. Similarly, levels of APOC3 were significantly higher in patients with malignant ovarian cancer in comparison to benign ovarian cyst samples [78]. Despite the limited sample size, the association with APOC3 reached statistical significance, with a *p*-value of 0.04. APOD is frequently linked with HDL in the plasma and is associated with favorable prognosis in cancer [79,80]. Concordantly, in epithelial ovarian carcinoma, the overall survival of patients was higher in APOD-positive tumors as compared to APOD-negative tumors [80]. Furthermore, APOD levels did not show a significant correlation with the presence of ovarian cancer to warrant its use as a diagnostic indicator, given that only 18 out of the 68 tested samples exhibited positive staining [80]. Yet, a notable correlation was found with prognosis for tumors that were larger than 1 cm in size. Contrary to the role of APOD in ovarian cancer, the overexpression of APOE was found in the sera of ovarian cancer patients in comparison to healthy individuals [81,82,83]; this overexpression was vital for the growth and survival of ovarian cancer [82]. Chen and colleagues [82] inhibited APOE expression in vitro and reported cell cycle arrest in the G2 phase as well as the induction of apoptosis, further supporting the role of APOE in ovarian cancer cell proliferation and survival. In addition, the upregulated expression of APOJ has been noted in the plasma of ovarian cancer patients and indicated as an early diagnostic and predictive marker for adverse outcomes [84]. A large-scale serial analysis in ovarian cancer tumors found that the APOJ gene was upregulated in malignant samples as compared to non-malignant samples [85]. The most recent analysis studied a multiple biomarker combination including APOA1 and APOA2 for ovarian cancer [86]. The results showed that the most optimal biomarker combination was a panel of five markers: CA 125, HE4, CA 15-3, APOA1, and APOA2, giving a sensitivity of 93.71% and a specificity of 93.63% for detecting ovarian cancer [86]. 

In cervical cancer research, APO expressions have been identified, mirroring the patterns observed in ovarian cancer. An investigation into APOA1 expression among cervical cancer patients, as compared to a control group, noted a significant decline in APOA1 levels in the patients, positioning it as a possible biomarker for cervical cancer [87]. However, this was an initial study with a small sample size, indicating that further investigation with a larger cohort is required for substantiation. Regarding post-treatment prognosis in cervical cancer, APOC2 might be a prospective marker, given the absence of a substantial difference between the control group and asymptomatic patients [88]. Conversely, in cases of cervical cancer leading to mortality, APOC2 levels were significantly lower than in asymptomatic cases [88]. The small scale of the study, which included only 9 controls and 28 cervical cancer cases, underscores the need for broader research efforts. Similarly, in the context of ovarian cancer, a separate study identified that genes associated with the pathogenesis of invasive cervical cancer, including the APOD genes, were downregulated in affected patients [89].

Research on APOs in endometrial cancer is sparse. One study investigated whether APOA and APOB could be considered as independent risk factors for lymphovascular space invasion in type 1 and type 2 endometrial cancer [90]. The findings indicated that APOB is an independent risk factor exclusively for type 1 endometrial cancer [90]. Additionally, a separate study examined APOD expression in endometrial cancer tissues and found that only 34% of the samples exhibited positive APOD expression, leading to the conclusion that there was no significant association [91]. Table 3a (ovarian cancer) and b (cervical cancer) provide a summary of the clinical impact, action, consequence, control, prognostic significance, and origin of various apolipoproteins in the context of gynecological cancers.

## 7. Apolipoproteins in Lung Cancer

Several studies have pointed towards a correlation between APOs and lung cancer. Studies have shown a loss of APOA1 and APOB expression in lung cancer patients [92]. The increased APOB/APOA1 ratio correlates with a higher incidence of lung cancer in both males and females [93,94]. This association was confirmed in another study which was more specific to small-cell lung cancer (SCLC) [95]. It was also found that the rate of oxidative stress is elevated in patients with a higher APOB/APOA1 ratio, adding to the speculation that these apolipoproteins can increase the incidence of lung cancer [94,95]. While APOA1 is recognized for its role in cardiovascular disease, an in vitro study carried out in mice proved that APOA1 has anti-tumorigenic roles [96]; mice with the human APOA1 transgene (A1Tg) experienced hindered cancer growth and progression [97]. However, the findings of this study are particular to APOA1 and should not be extended to other members of the APOA protein family, including APOA2.

The underlying mechanism responsible for the anti-tumorigenic role of APOA1 is that APOA1 can convert cancer-linked macrophages from being pro-tumor M2 to becoming the anti-tumor M1 phenotype [97]. APOA1 has been shown to reduce lung cancer through its immunomodulatory mechanisms and anti-inflammatory characteristics by inhibiting the neo-angiogenesis of lung tumors while also reducing enzymes that enable cancer metastasis [75,98]. Contrary to previous findings regarding APOA1 expression in lung cancer, a recent study found that APOA1 levels were increased in patients with idiopathic pulmonary fibrosis-related lung cancer, resulting in dyslipidemia [99]. Another study determined that APOA1 levels were significantly elevated in SCLC, despite not being markedly increased in non-small-cell lung cancer (NSCLC), and found that reduced APOA1 levels correlated with an increased recurrence of SCLC [100]. These contrasting findings highlights the need for a nuanced approach to researching APOA1 in lung cancer. It is evident that APOA1’s role is not uniform across different lung cancer contexts and that its function may be influenced by the underlying pathology of the lung condition. Future research must consider these subtleties to fully elucidate APOA1’s potential as a biomarker and therapeutic agent in lung cancer.

While there is a substantial number of studies exploring the function of APOA1 in relation to lung cancer, research on the role APOA2 in lung cancer is scarce. Nevertheless, a study was conducted to assess tumor and inflammatory markers and their role in diagnosing lung cancer, such as APOA2, an inflammatory marker that was significantly differentiated between non-small-cell lung cancer (NSCLC) patients and controls, with a sensitivity of 89% [101]. When combined with other inflammatory markers and tumor markers, APOA2 was successful in diagnosing early-stage lung cancer patients, including NSCLC [101]. This observation applied to SCLC as well; however, the use of APOA2 as an early diagnostic marker requires further validation. On the other hand, the role of APOA4 in lung cancer is contradictory depending on the lung cancer subtype. While enhanced APOA4 expression was reported in squamous cell carcinomas of the lung [102], in adenocarcinomas, a loss of APOA4 was reported in serum [103], which is comparable with previous findings from similar studies. These contrasting findings around APOA2 and APOA4 within lung cancer subtypes are a testament to the molecular diversity of the disease. They reinforce the need for a stratified approach in cancer research, where the nuances of each subtype can significantly influence the course of diagnosis and treatment. Moreover, these insights into APOA proteins could help to unravel the broader complexities of cancer pathophysiology, potentially guiding targeted therapies and precision medicine. It is a growing area of research that promises to refine our understanding of cancer biology and improve patient outcomes through more personalized diagnostic tools.

The role of APOB is controversial in lung cancer. As previously stated, studies pointed towards the increased incidence of lung cancer associated with increased APOB serum levels [93]. While this study is notable for its extensive sample size, it does carry a set of limitations. These limitations include the potential for reverse causation bias, the lack of adjustments for lipid-lowering medications, which could ultimately impact apolipoprotein levels, and a cohort of participants that may not be entirely representative of the broader population. However, other studies showed that the downregulation of APOB was correlated with an increased risk of cancer [94,104]. In addition, it was found that APOB has varying genotype expression in different cancers, with some genotypes associated with favorable outcomes and others resulting in inferior survival rates in NSCLC [104]. While the mechanism that correlates APOB and NSCLC is unclear, it is hypothesized that the association is due to APOB’s role in regulating cholesterol transport and metabolism, which modulate the development of NSCLC [104]. 

Much like the in research that was carried out on the role of APOA1 in SCLC, the expression of APOC3 was significantly decreased in SCLC patients as compared to NSCLC patients and normal lung tissues [100]. The significant loss of APOC3 expression in SCLC tissues suggests the use of APOC3 as a predictive marker for SCLC [100]. Additionally, the expression of APOC3 was remarkably elevated in patients with recurrence [100]. 

APOE, known to be implicated in cardiovascular and neurological diseases, is involved in tumorigenesis, cancer cell proliferation, and metastasis, and is associated with amplified oxidative stress [105,106]. However, its role in lung cancer remains unclear. In one study, APOE was found to be upregulated in patients diagnosed with NSCLC by 1.6-fold; however, the use of APOE as a candidate biomarker remains insignificant [107]. This finding was supported by an in vivo study in which APOE knockdown inhibited the proliferation and metastasis of lung cancer cells [108]. In addition to the correlation between APOE upregulation and increased lung adenocarcinoma frequency, APOE is associated with a higher incidence of malignant pleural effusions (MPEs), a complication of lung adenocarcinoma, compared to lung cancers without MPEs [109]. 

APOH is yet another protein with inflammatory effects. A correlation between increased APOH and NSCLC has been established; however, the underlying mechanism is still nascent [110]. Likewise, APOH was upregulated in papillary lung adenocarcinomas in mice; however, in mice with atypical adenomatous hyperplasia (AAH) of the lung, their APOH was downregulated by twofold. It was discovered that APOH inhibits angiogenesis by suppressing endothelial cell growth. It is therefore suggested that its downregulation in AAH limits its inhibitory effects on angiogenesis, and this plays a role in promoting cancer proliferation [103,111]. The research additionally discovered that APOM expression is suppressed in the AAH tissue [103]. The significance of this finding is due to the role of APOM as a primary carrier for sphingosine-1-phosphate, which is a signaling molecule responsible for inhibiting ceramide; the inhibition of ceramide leads to suppressed apoptosis and increased cell proliferation, leading to cancer [72]. Therefore, blocking the inhibitor of ceramide would ensure that ceramide retains its apoptotic effects, thus reducing the risk of tumor development [103]. Therefore, APOM could potentially play a role in a tumor-suppressing mechanism, yet this requires further studying [103]. In contrast to previous studies, a recent study by Zhu and colleagues [112] reported that the upregulation of APOM stimulates cell proliferation, invasion, and tumor development in NSCLC by inducing sphingosine-1-phosphate, leading to the activation of the ERK1/2 and PI3K/AKT signaling pathways.

The function of APOL and its subtypes remains understudied in the current literature. It was established that APOL2 lacks apoptotic potential [113]; however, in human bronchial epithelium, APOL2 has exhibited anti-apoptotic ability [114]. In human lung cancer tissue, APOL2 expression was shown to be augmented [114]. Table 4 concisely outlines the roles, regulatory mechanisms, clinical effects, prognostic value, and sources of different apolipoproteins in lung cancer.

## 8. Apolipoproteins in Colorectal Cancer (CRC)

Colorectal cancer (CRC) is a complex and heterogeneous disease that arises from the accumulation of genetic and epigenetic alterations in colon epithelial cells. Despite advances in diagnosis and treatment, CRC remains a leading cause of cancer-related death worldwide. Recent studies have suggested that apolipoproteins may also have significant functions in the development and progression of CRC. 

A recent retrospective study highlights the multifaceted role of serum APOA-I in colorectal cancer (CRC). The link between reduced serum APOA-I levels and larger tumor sizes, along with more advanced TNM stages [30], is indicative of a more extensive cancer spread, which points to APOA-I’s potential involvement in the disease’s progression. Furthermore, the association with biomarkers of systemic inflammation [30] underscores the complexity of APOA-I’s function, suggesting that it may influence both lipid metabolism and inflammatory pathways in cancer development and progression. These insights suggest that serum APOA-I levels could serve as a biomarker reflecting various aspects of CRC pathogenesis, including lipid dysregulation and inflammation, and might be relevant for understanding the mechanisms of cancer development, offering potential prognostic value and possibly informing therapeutic strategies. Nevertheless, this study was limited to patients from single cancer center, and all the patients were Chinese, so it has selection and sampling bias. Therefore, the conclusion may not be suitable for extrapolation to Western populations due to a lack of internal validity. A different study using a prospective cohort study design assessed the serum lipoprotein levels at baseline and then over repeated assessments that explored the correlation between different apolipoprotein concentrations and tumor subsites of colorectal cancer (CRC) patients; it was observed that there is no statistical significance between the concentration of APOA and early-, middle-, or late-onset CRC. Additionally, a negative correlation between APOA and cancer in the hepatic flexure was found, with high APOA associated with a lower risk of hepatic flexure cancer. However, this study was limited to its use of the baseline concentration for its main analysis, and therefore, a short-term variation subsample assessment indicated that time-dependent variation in lipids was unlikely to have a substantial impact on the results [115]. These findings suggest a possible anti-tumor impact of APOA, but further research is required to determine the underlying mechanisms. Another recent study reported that ApoA1 and ApoA1-binding protein suppress colorectal cancer (CRC) cell proliferation and metastasis, creating a synergistic effect against CRC migration and angiogenesis by increasing cholesterol efflux and damaging the correct distribution of invasion [116]. Therefore, increased APOA levels are a favorable factor in metastatic colorectal cancer (mCRC) for overall survival. Therefore, APOA mimetic peptides are proposed as therapeutic molecules that mimic APOA’s structure and function [7]. APOA mimetic peptide 4F (L-4F) exerts an anti-inflammatory effect by reducing levels of tumor necrosis factor (TNF-α) and interleukin-6, which have been shown to Inhibit cancer development both in vitro and in vivo [31].

Studies have also highlighted the significance of APOB in colorectal cancer (CRC) development. Yang et al. reported a positive correlation between high circulating APOB and CRC risk, particularly in men, which could be linked to APOB’s role as a lipid carrier for cholesterol and triglycerides into extrahepatic tissues [30]. Moreover, it was found that the glycated APOB form was more prevalent in CRC and adenoma tissues than non-cancerous tissues, suggesting a potential role for APOB in dysplastic and neoplastic development [116]. However, the role of APOB in CRC development is still controversial. It was proposed that APOB might be downregulated in tumors due to inactivating mutations in the APOB gene [31]. As APOB synthesis and secretion require abundant energy, tumor cells may conserve energy for their proliferation by inactivating the APOB gene. Other studies have even identified no association between APOB and tumor stage, which further complicates the role of APOB in CRC development.

Nonetheless, APOB has found a medical application as a predictor of survival in CRC patients after radical surgery, predicting patient outcomes, and could potentially be used as a therapeutic target [116]. 

Other findings have revealed that an elevated APOB/APOA1 ratio was associated with worse survival in mCRC patients and was identified as an independent prognostic factor for overall survival in mCRC, with higher levels causing shorter overall survival. The proposed explanation is that ApoA-I is negatively associated with tumor-induced systemic inflammation, while elevated ApoB indicates a higher systemic inflammatory marker, and so a high ApoB/ApoA-I ratio, which is considered atherogenic, may contribute to tumor necrosis. However, this is merely a proposal; the precise process was not detailed in the study [30]. Although, another study which assessed this prognostic factor found that a high APOB/APOA1 ratio was found to predict poorer survival in patients with metastatic CRC to the liver, as well as in patients with advanced rectal cancer [116]. 

APOC1 is a secretory protein that is commonly associated with very-low-density lipoproteins (VLDLs) and low-density lipoproteins (LDLs). APOC1 promotes cell proliferation and migration in CRC through the P38-MAPK signaling pathway. Specifically, APOC1 promotes the cell phosphorylation of P38, leading to an increased capacity for the G2/M phase and decreased cells in the G0/G1 phase [117]. These results suggest that APOC1 promotes the progression of the cell cycle and may serve as a predictive marker for clinicopathological significant events in CRC. The underlying mechanism for this effect remains unknown, and more research is needed to fully elucidate the role of APOC1 in CRC and its potential use as a diagnostic or prognostic marker. According to additional research, APOC1 overexpression aids in the progression of liver metastases in colorectal cancer [116].

The behavior of APOD is highly unusual. The downregulation of APOD mRNA expression, caused by DNA methylation to its promoter gene, correlates with decreased protein expression [116]. One phenomenon that has been observed is its ability to react to oxidative stress that is exacerbated by an increase in the concentration of reactive oxygen species (ROS), which is commonly observed as cancer progresses through its stages, and this is positively correlated with the APOD concentration. The conundrum it raises is that, although APOD is inversely connected with tumor advancement, its action is the opposite of the ROS function, which has been shown to cause cancer to advance. To test this, a paraquat-triggered oxidative stress condition was created, and the exogenous introduction of APOD to CRC cells enhanced tumor suppression through apoptosis [118]. Furthermore, the downregulation of APOD is linked to lymph node metastasis, advanced disease stages, and a worse prognosis [7]. According to a recent study, a reduction in APOD levels was linked to the initial stages of cancer development, specifically stages I and II, but not to later stages [118]. Hence, the findings suggest that APOD levels can be utilized as an early diagnostic marker for cancer initiation rather than a marker for tumor progression after initiation.

APOE function has been seen to be highly controversial between studies. Whilst some studies associate APOE with tumor progression, another study proposes APOE as a potential protective factor. The study which proposed APOE as a protective factor associated the APOE gene being silenced with an increased susceptibility to inflammation-related tumorigenesis [116]. On the other hand, a different study suggested that APOE activation was seen to restrict the immune system suppression of cancer cell proliferation, thus promoting cancer growth and metastasis. Nevertheless, these studies all showed a significant APOE level upregulation in CRC. Further patterns have also been identified, like APOE levels being significantly higher when a tumor has metastasized to the liver. One study specified that this upregulation occurs only in primary CRC, not in stage II of CRC [116]. A murine model using wild-type mice showed that APOE upregulation was also associated with enlarged tumor sizes. The proposed mechanisms through which APOE accelerated cancer is in relation with intracellular adhesion and junctions, thereby decreasing cell contact inhibition and polarizing normal cells to tumor cells through the PI3K/Akt/mTOR pathway [118]. APOE’s polymorphism has also been studied. There are three different APOE alleles—APOE-ε2 (cys112, cys158), APOE-ε3 (cys112, arg158), and APOE-ε4 (arg112, arg158)—that differ only due to two amino acids. APOE-ε4 is associated with reduced proximal colorectal neoplasia in the forms of adenoma and carcinoma; however, investigation into distal neoplasms have shown no significant difference [31]. Yet these findings still need further investigation, as another study involving Japanese males could not identify these correlations between APOE-ε4 and proximal adenomas, suggesting that APOE is affected by other factors other than genes, for example, ethnicity in this case. Other factors that may determine the potential association between the genotypes of APOE and colonic cancer include racial variation, genetic background, diet, and physical training, which have likely led to a discrepancy in findings on the carcinogenicity of the allele ε4. APOE-ε3 has also shown an inverse correlation between concentration and colon cancer; a deficiency in APOE-ε3 leads to colon cancer, which has been especially observed populations over 50 years of age [116]. APOE serum levels have been proposed as a diagnostic marker for metastatic CRC under chemotherapy and bevacizumab treatment. However, further research is needed to fully understand the potential of APOE as a biomarker for clinical outcomes. 

In one study, the high expression of APOH in one group led to a worse prognosis compared to the low-expression group. Despite this finding, the underlying mechanisms of APOH in CRC remains unknown and requires further investigation [119]. 

Clusterin, also known as apolipoprotein J (APOJ) is a heterodimeric glycoprotein that is essential for clearing away dead cells and apoptosis. APOJ is significantly increased in colon cancer and contributes to multistage colorectal carcinogenesis. Additionally, research has demonstrated that APOJ stimulates colon cancer metastasis and tumor invasion via the p38/MAPK/MMP9 pathway. It has been discovered that APOJ is a cytoprotective chaperone protein that promotes the folding of released proteins and can be activated by stress. Its three isoforms take part in both pro- and anti-apoptotic activities [7]. However, despite its significant role in CRC, the mechanisms underlying the pro- and anti-apoptotic activities of APOJ in CRC remain largely unknown.

Studies have proposed conflicting findings regarding APOM expression levels and its underlying mechanisms in CRC. It was reported that APOM expression was significantly reduced in CRC tissue compared to adjacent normal tissue [7]. The authors further investigated the potential role of APOM in regulating the epithelial–mesenchymal transition (EMT), a process involved in tumor invasion and metastasis. The study found that the overexpression of APOM in CRC cells inhibited EMT by decreasing the expression of EMT-related transcription factors and matrix metalloproteinases (MMPs). These results suggested a potential tumor suppressor role for APOM in CRC through the inhibition of EMT. In contrast, a study investigating the effect of APOM on CRC cell proliferation and apoptosis found that the upregulation of APOM was associated with lower apoptosis rates and higher tumor growth in Caco-2 cells [120]. The suggested mechanism includes the increased expression of ribosomal protein S27A (RPS27A), which has been observed to promote cell growth and invasion, regulate the cell cycle, and impede programmed cell death via various pathways in both living organisms and lab conditions [120]. According to certain reports, RPS27A can interact with the mouse double minute 2 (MDM2) gene, which is a primary negative regulatory factor of the p53 protein. This interaction suppresses MDM2, which, in turn, results in the activation of p53, inducing cell cycle arrest [120]. In response to ribosomal stress, MDM2 ubiquitinates RPS27A, leading to its proteasomal degradation, thus creating a mutual-regulatory loop. These findings suggest that APOM may act as an oncogene in CRC through the RPS27A-MDM2-p53 pathway. These contrasting findings suggest a controversial role for APOM in CRC tumorigenesis and progression. APOM expression levels and its underlying mechanisms may differ depending on the stage and grade of CRC. Further investigation is needed to fully elucidate the role of APOM in CRC and its potential use as a therapeutic target. Table 5 details the functions, regulatory pathways, clinical outcomes, prognostic importance, and origins of various apolipoproteins in colorectal cancer.

## 9. Apolipoproteins in Pancreatic Cancer

The literature suggests that different subtypes of APOA play a role in the pathogenesis and potential treatment of pancreatic cancer. APOA1 was found to be highly expressed in tumor tissue compared to non-tumor tissue, suggesting its potential use as a sensitive and specific marker for early-stage pancreatic neoplasms [31]. Using advanced mass spectrometry-based techniques, it was shown that using a panel of APOA1, APOE, APOL1, and trypsin inhibitor heavy chain H3 (ITIH3) can provide a sensitivity of 95% and specificity of 94.1% in the diagnosis of pancreatic cancer [1]. In addition, it was found that low levels of APOA1 can be used to differentiate type 2 diabetes secondary to pancreatic cancer from common type 2 diabetes mellitus [31]. A study used mass spectrometry analysis to identify TRIM 15, an E3 ubiquitin ligase, as a potential target for treatment that is a binding partner for APOA1 [121,122]. Downregulating the expression of TRIM15 led to increasing the levels of APOA1, which was found to suppress the metastasis of pancreatic cancer. However, this study was focused on genetic and biochemical analysis, and it did not address the potential limitations of the downregulation of TRIM 15, considering its important roles in tumor suppression and cell apoptosis.

Patients with pancreatic cancer were found to have a decrease in the plasma concentration of a specific isoform of APOA2 called APOA2-ATQ/-AT [123]. A study investigated the effect of chemoradiotherapy (CRT) on the levels of APOA2 and revealed that the distribution of APOA2 isoforms in pancreatic ductal adenocarcinoma (PDAC) patients underwent significant changes before and after CRT, which were not linked to the treatment effectiveness, but rather to alterations in the pancreas morphological characteristics. APOA2-ATQ/AT was deemed to potentially be a useful marker for detecting PDAC, but not for the efficacy of CRT at different stages of PDAC [124]. The findings of a retrospective study indicate that low serum levels of APOA2-ATQ/AT can be a potential biomarker for identifying intraductal papillary mucinous neoplasm (IPMN) patients at high risk of developing PDAC. The study further showed that APOA2-ATQ/AT is more sensitive than the commonly used CA 19-9 serum marker in detecting patients with potentially curable IPMNs [125]. Prospective studies should be performed to validate these findings, especially since there were discrepancies in age among the diseased and control groups. Another study showed that diabetic patients with pancreatic cancer had significantly lower plasma levels of APOA4 compared to cancer-free diabetic patients. In addition to that, it was reported that the expression of APOA4 RNA did not differ significantly among the various stages of pancreatic cancer. Nonetheless, higher levels of APOA4 in the tumor tissue appeared to be associated with lower overall survival [126]. The mechanism of reduction of APOA4 expression in diabetes patients with pancreatic cancer was not investigated in this study. Furthermore, no justification was provided for the reduced levels of APOA4 in this subset of patients with diabetes. Further research is required to address these gaps, and to explore the relationship between carcinogenesis and APOA4 in non-diabetic pancreatic cancer patients.

While the relationship between APOB and pancreatic cancer has not been studied, the APOB mRNA editing catalytic subunit (APOBEC3C) was found to be the most expressed APOBEC enzyme in PDAC. Several studies found that higher levels of APOBEC3C expression were associated with shorter overall survival in PDAC patients. This is because high levels of APOBEC3C expression can result in focal hypermutation and increased tumor plasticity, making tumors more adaptable to chemotherapy and other evolutionary pressures, which increases the likelihood of developing new phenotypes and leads to worse outcomes for patients [127]. 

Earlier research has identified that individuals with a neoplastic pancreatic epithelium have heightened levels of APOC1 expression [7]. APOC2 has been shown to enhance cell growth and invasion in pancreatic cancer cell lines, indicating its potential as a predictor for cell survival [31]. Further research is needed to investigate the role of APOC in the pathogenesis and therapy of pancreatic cancer.

APOE was found to contribute to immunosuppression and inhibiting the apoptosis of malignant cells in pancreatic cancer. APOE contributes to immune suppression in pancreatic cancer by activating the NF-κB signaling pathway. This triggers the production of CXCL1, which is a protein that plays a role in immune suppression by recruiting immune cells that hinder the immune response to the tumor. It was concluded that blocking the production of CXCL1 may reverse APOE-mediated immune suppression. Furthermore, APOE expression was found to be higher in pancreatic cancer tissue than in normal pancreatic tissue, and that elevated APOE levels are linked to worse outcomes for pancreatic cancer patients [128,129]. In addition, APOE2 has been found to aid pancreatic cancer cells in avoiding mitochondrial apoptosis by regulating the mitochondrial localization and expression of BCL-2 through the activation of the ERK1/2/CREB signaling cascade [130]. A study found that pancreatic cancer cell proliferation is associated with increased expression of APOE2-LRP8, a ligand–receptor pair that promotes tumor progression. The APOE2-LRP8 axis appears to be a dominant biological cascade in this process, inducing the expression of p-ERK1/2 and c-Myc, both of which are involved in the cell cycle and promote tumor growth. The study suggests that targeting the APOE2-LRP8 axis could be a promising therapeutic strategy for pancreatic cancer [131]. These findings suggest that ApoE2 could be a useful prognostic marker and a potential target for developing novel therapies against pancreatic cancer, yet further research is needed. 

APOJ, or Clusterin (CLU), is a well-studied apolipoprotein in the context of pancreatic cancer. Multiple intracellular proteins control the expression of CLU at either the mRNA or protein level, either directly or indirectly, to manage cell growth and proliferation [132]. A study highlighted that CLU expression in pancreatic cancer is regulated by HSF1, a stress-induced master regulator that is known to play a key role in converting fibroblasts into cancer-associated fibroblasts (CAFs) in pancreatic cancer and other cancers [133]. CLU participates in the modulation of various signaling pathways related to cell proliferation, such as ERK, AKT, and NF-κB, and is also involved in receiving and interpreting extracellular signals. A reduction in the level of CLU can lead to cell proliferation, epithelial-to-mesenchymal transition (EMT), and decreased sensitivity to gemcitabine chemotherapy, ultimately resulting in the progression of the disease and poor prognosis [134]. Additionally, CLU is linked to an early resistance to MEK inhibitors, a type of cancer treatment targeting the mitogen-activated protein kinase (MAPK) signaling pathway. Therefore, CLU could play a crucial role in the development of a novel therapeutic approach for PDAC [135].

APOL1 appears to play a complex dual role in the development and progression of pancreatic cancer. In vitro, the inhibition of APOL1 significantly reduced cell growth and caused cell cycle arrest and apoptosis, while also decreasing cell proliferation in vivo, as demonstrated by the smaller tumor size in a pancreatic cancer mouse model [136]. However, APOL1 also inhibits cell proliferation by activating the NOTCH1 signaling pathway. When activated, NOTCH1 induces the expression of several genes that are involved in regulating the cell cycle and promoting apoptosis, which in turn inhibits cell proliferation. Therefore, the ability of APOL1 to activate NOTCH1 signaling is thought to play a critical role in regulating cell proliferation and apoptosis in pancreatic cancer cells [136]. Given that this study was performed on mouse models, further research using human cells is required to investigate the effects of APOL1 in the pathogenesis of pancreatic cancer and to address its potential as a biomarker. In addition to that, there is controversy about whether APOL1 is upregulated or downregulated in pancreatic cancer, and more research is needed. Table 6 efficiently summarizes the roles, control mechanisms, clinical impacts, prognostic relevance, and derivation of different apolipoproteins in pancreatic cancer.

## 10. Apolipoproteins in Hepatic Cancer

APOA, specifically APOA1, has been highlighted as an important diagnostic biomarker of hepatic cancer. It has been discovered that APOA1 can be used to differentiate between individuals who are healthy and those who have liver disease, particularly cirrhosis and hepatocellular carcinoma (HCC) [137]. It was found that APOA1 along with other proteins including ISY1, SYNE1, MTG1, and MMP10 are highly expressed during the early stages of HCC. These proteins form a network that interacts with key regulators of lipid metabolism as well as splicing pathways, suggesting their potential role in the development of HCC [138]. In a study conducted on patients with HCC undergoing trans-arterial chemoembolization (TACE), the neutrophil-to-APOA1 ratio (NAR) was identified as an independent predictor of overall survival. NAR was found to be indicative of the levels of lectin-type oxidized low-density lipoprotein receptor-1-positive polymorphonuclear myeloid-derived suppressor cells in circulation (LOX-1 + PMN-MDSCs), which are immune cells that can suppress the immune response and promote cancer progression [139]. APOA1 is therefore a potential target for treatment. It has been indicated that ellagic acid, a polyphenol present in nuts and fruits, could potentially have therapeutic effects in reducing the risk of hepatic cancer and cardiac disease by regulating the levels of APOA1 [140]. While this has been investigated in vitro, the detailed mechanism underlying the effects of ellagic acid on lipoprotein metabolism was not addressed, suggesting a need for further research. 

Hepatic cancer and liver metastasis have been found to be associated with increased levels of APOB, which indicate its potential significance in tumorigenesis and disease management. The development of HCC has been linked to mutations in the APOB gene. A truncation of the APOB protein resulting from the mutation may increase the risk of HCC, especially in hypocholesterolemia patients [31]. It was found that certain single nucleotide polymorphisms (SNPs) in the APOBEC3 gene family were associated with the progression of chronic hepatitis B and the development of HCC in a Chinese population [141]. Moreover, the expression of APOBEC3G was significantly higher in liver metastases than in primary liver tumors or non-cancerous liver tissue, suggesting that APOBEC3G could be a potential biomarker for predicting liver tumor metastasis [31]. A study found that the APOB/APOA1 ratio is a specific predictor of liver metastasis in rectal cancer patients [142]. Nonetheless, this study was retrospective and only included patients with locally advanced rectal cancer who received chemoradiotherapy followed by surgery, which may not represent the entire population of rectal cancer patients. APOB was also highlighted as a useful tool to predict resistance to treatment, where serum lipid levels of APOB as well as APOA1 were found to be useful in predicting the response of patients with advanced intrahepatic cholangiocarcinoma to PD-1 inhibitor treatment [143]. 

The prognostic value of APOC has been investigated in multiple studies. A study found that APOC1, APOC2, APOC3, and APOC4 were expressed differently in tumor and non-tumor tissues in hepatocellular carcinoma. APOC1 and APOC4 were associated with overall survival, and APOC3 was associated with both overall survival and recurrence-free survival [144,145,146]. A progressive increase was observed in the expression of APOC1 from normal tissue to primary tumor tissues and liver metastatic tumor tissues in colorectal cancer, suggesting that APOC1 could have a significant role in the pathology of liver metastasis in colorectal cancer. It is possible that it can trigger the transformation of tumor-associated macrophages (TAMs) into M2-like cells, which can subsequently contribute to immune evasion and angiogenesis in tumor cells [146,147].

APOJ has been found to be a more effective biomarker for diagnosing HCC compared to other used markers, including alpha-fetoprotein (AFP), pCEA, and CD10. When combined with AFP, APOJ further improves diagnostic accuracy. Furthermore, APOJ performed better than both pCEA and CD10 in differentiating liver malignancies from benign hepatocellular masses [148,149]. Regarding APOJ, it has also been found that the combination of filamin-A and APOJ genes could potentially serve as a useful marker for hepatocellular carcinoma [150]. Nonetheless, this has to be investigated in vitro to determine whether an elevated expression of filamin-A and APOJ is specific to HCC. In terms of its role in pathogenesis, it was suggested that APOJ may play a role in advancing the progression of HCV-related HCC by modulating autophagy, and therefore, could be a promising target for treatment [151]. It was shown that reducing the levels of APOJ in the bloodstream resulted in reducing resistance to treatments such as sorafenib/doxorubicin, while increasing the levels of APOJ leads to increasing metastasis and tumor growth [152]. Table 7 effectively encapsulates the functions, regulatory processes, clinical consequences, prognostic significance, and sources of various apolipoproteins in hepatic cancer.

## 11. Apolipoproteins in Prostate Cancer

APOA1 expression is upregulated in prostate cancer. APOA1 is regulated by MYC, a frequently amplified oncogene in late stages. Therefore, it can predict prognosis and recurrence. Patients at risk of metastasis or neuroendocrine prostate cancer would benefit from this. Since its expression increases with disease progression, it is suggested that the source of APOA1 is the tumor cells [153]. Similarly, APOA2 was proven to be overexpressed in prostate cancer, specifically the 8.9-kDa isoform of APOA2 [154]. One study reported an increase in serum APOC1 protein levels in patients during disease progression, suggesting an association with prostate cancer progression. However, the exact role of APOC1 in prostate cancer pathogenesis remains unclear. An immunohisto-chemical analysis revealed that APOC1 was predominantly found in the cytoplasm of hormone-refractory cancer cells [155]. APOC1 mediates the cell survival, cell cycle distribution, and apoptosis of prostate cancer via activating the survivin/Rb/p21/caspase-3 signaling pathway [156]. The malignant transformation of the prostate is associated with an increased expression of APOD. APOD immunoreactivity was observed in areas of high-grade prostatic intraepithelial neoplasia (HGPIN) in 82% of prostatectomy specimens. The expression of APOD in HGPIN suggests its potential role as a cellular marker for HGPIN and prostate cancer [157]. Prostate tumor cells secrete increased amounts of APOE, which binds to TREM2 on neutrophils, inducing senescence. The increased expression of APOE and TREM2 in prostate cancer correlates with poor prognosis. APOE is believed to be produced by prostate tumor cells [158]. Certain single-nucleotide polymorphisms in the APOL3 region on chromosome 22q12 increased susceptibility to hereditary prostate cancer [69]. High levels of APOJ are found in prostate cancer, correlating with tumor grade, and potentially contributing to treatment resistance. Therefore, small interfering RNA (siRNA) oligonucleotides targeting APOJ silence its gene expression, resulting in a significant increase in chemosensitivity [159]. Table 8 provides a compact summary of the activities, control systems, clinical effects, prognostic value, and origins of different apolipoproteins in prostate cancer.

## 12. Apolipoproteins in Gastric Cancer

APOA levels in gastric cancer are controversial. Some studies have shown that APOA levels become elevated, especially in the early stages of gastric cancer. A mouse model study showed that high levels of circulating APOA-1 were associated with an increased tumor burden. Post-gastrectomy, their APOA levels dramatically decreased; hence, it is believed that APOA-1 is secreted by the tumor cells. Further research is needed to understand the specific role and mechanism of APOA-I in gastric cancer cells [160]. However, one study suggested that APOA could distinguish between chronic gastritis and gastric cancer; APOA levels were found to be higher than normal in chronic gastritis and lower than normal in gastric cancer [161].

APOA2 can also be used as a prognostic indicator for gastric cancers high in Clau-din-6. The APOA2 gene is highly expressed in gastric cancers with high Claudin-6 levels, affecting cholesterol metabolism. Hence, APOA2 is suggested to be an effective prognostic marker for such cancers [162]. 

Regarding APOC, contradictory results have been found on its role in gastric cancer. Some studies have shown that lower APOC1 and APOC3 levels signify a poorer prognosis [163,164]. Other studies claim that APOC1 is overexpressed in gastric cancer [165]. In gastric cancer patients with peritoneal metastasis, APOC2 was over-expressed and was associated with poor prognosis. This is because APOC2 promotes the CD36-mediated PI3K/AKT/mTOR signaling pathway. The over-activation of mTOR increases cell survival and cell cycle progression. The inhibition of APOC2 has been shown to delay tumor progression [166]. In one study, it was shown that APOA2, APOC1, and fibrinogen a-chain were distinguishing biomarkers that could diagnose gastric cancer. The sources of these biomarkers are unknown [167]. Another study demonstrated that zinc finger protein 460 can promote APOC1 transcription, accelerating the epithelial–mesenchymal transition (EMT) and the development of gastric cancer [168].

APOE upregulation was associated with shorter survival times for people with gastric cancer. Increased APOE levels are strongly associated with the risk of muscular invasion. Therefore, it could be used to predict gastric tumor invasion [169,170]. In gastric cancers, the overexpression of APOJ is associated with tumor progression and metastasis [171]. Table 9 offers a brief encapsulation of the roles, regulatory frameworks, clinical outcomes, prognostic implications, and sources of various apolipoproteins in gastric cancer.

## 13. Apolipoproteins in Thyroid Cancer

Thyroid cancer is associated with apolipoproteins. APOA is a well-documented apolipoprotein in thyroid cancer. APOA1, APOA2, and APOA4 were found to be downregulated in female patients with thyroid cancer [172,173]. However, APOA1 was greater in the subset of patients with papillary thyroid cancer (PTC) metastasis [172]. APOA1 was found to be one of the three most vital genes regulating the lipid proteomic profiles in humans [172]. Similarly, APOA1 and APO4 were implicated in the LXR/RXR activation pathway, which mediates cholesterol metabolism and excretion [172,174]. As such, its reduced levels in thyroid cancer lead to a dysregulated lipid profile [172]. This dysregulated lipid profile is said to alter gut microbiota symbiosis, which increases the risk of thyroid cancer progression [175]. Nevertheless, the way in which lipid breakdown is significant in cancer remains understudied and requires further research to facilitate the development of new cancer therapeutics [172]. One study suggested that HDL-C has anti-inflammatory and antioxidant properties, mediated mostly by APOA1, which could play a possible role in cancer mediation [176]. Further studies confirmed that reduced APOA1 levels were associated with a worse prognosis and aggressive thyroid tumor characteristics [176]. Nevertheless, the previous findings contradict with a recent study that found no association between APOA1, APOB, or APOB/APOA1 levels and the risk of thyroid cancer [177]. In fact, it was found that elevated HDL-C and cholesterol levels, which are regulated by APOA1, were associated with a reduced thyroid cancer risk [177]. Other contradicting findings stated that APOA1 and APOA4 were overexpressed in PTC, the most prevalent subtype of thyroid cancer, when a proteome analysis was conducted on PTC cells [178,179].

A recent study found that APOA1 had an inverse relationship with tumor size in male PTC patients, especially in the younger age group after adjusting for age [180]. This study claimed that there was no association between rate of lymph node metastasis in PTC patients and serum APOA1 [180]. With regard to medullary thyroid carcinoma (MTC), it was discovered that APOA4 expression is increased, possibly identifying it as a biomarker for MTC diagnosis [181]. Moreover, in patients undergoing management for differentiated thyroid cancer (DTC), their APOA1/2 ratio levels remained increased in an evident hypothyroid state post-RAI therapy [182]. While APOA2 itself did not show any statistically significant findings in the hypothyroid state following treatment, the APOA1 levels decreased [182]. It is important to note that APOA1 and APOA2 returned to their baseline levels with levothyroxine therapy [182]. This concludes that in DTC, alterations in thyroid hormones are associated with the modulation of plasma levels of apolipoproteins. 

The association between cholesterol levels and thyroid cancer has also been studied [183]. It was found that serum LDL cholesterol and APOB levels were significantly lower in patients with more aggressive tumors [183]. APOB levels were also lower in patients with high-risk PTC tumors, as well as in individuals with poorly differentiated thyroid cancer (PDTC) and anaplastic thyroid cancer (ATC) [183]. This suggests that the role of lipids in aggressive thyroid cancer progression is mediated through APOB. Regardless of the previous findings, another study established that APOB is not associated with tumor size or the rate of lymph node metastasis in male PTC patients [180].

When potential biomarkers were investigated in relation to PTC through a support vector machine, APOC1 and APOC3 were found to be downregulated [184,185]. Through further investigation, it was discovered that these two APOs decrease as the cancer stage increased, further proving their downregulation in PTC. This finding could be utilized as a method of non-invasive diagnosis and staging for PTC in patients. The proposed pathway through which APOC1 and APOC3 are downregulated is through orphan nuclear hormone receptor superfamily receptors [184]. These orphan members bind to hormone response elements (HREs) and dramatically augment or suppress APOC3 activity [184]. Other studies have discovered that retinoid X receptor alpha (RXRalpha) and thyroid hormone receptor beta (T3Rbeta) can inhibit APOC3 when T3 is present [184]. On the other hand, it was established that liver X receptor β (LXRβ) was overexpressed in thyroid cancer [186]. The significance of this finding in relation to apolipoproteins is that APOC1 and APOC2 are transcriptional targets genes of LXRβ [186]. Therefore, the upregulation of LXRβ in thyroid cancer results in an increased expression of APOC1 and APOC2. When compared with normal cell lines, APOC1 showed a statistically significant overexpression, validating the previously mentioned findings [186]. 

While the association between APOD and thyroid cancer is understudied, one study found that APOD was downregulated in DTC [187]. In addition, APOD was associated with a higher risk score and was discovered to be a harmful factor that correlates with the recurrence of DTC [187]. The effect of its downregulation is perhaps related to APOD’s regulation by the P53 tumor suppressor family. As such, it is suggested that a decrease in APOD levels increases tumor proliferation, as proven by its ability to suppress tumor growth in other cancers such as breast, prostate, and colorectal cancers [187].

APOE is another well-documented apolipoprotein in thyroid cancer. It has been found that APOE expression is upregulated in thyroid cancer [181,186,188,189,190,191,192,193]. APOE gene expression was analyzed and revealed that it was significantly overexpressed in thyroid carcinomas, notably PTC [188,189,191,192]. Several databases validated this finding, including the TIMER, GEPIA, and Oncomine databases [188,189]. In normal cell lines, immunohistochemistry staining fails to identify APOE. However, the human protein atlas database proved that immunohistochemistry staining was able to detect sufficient levels of APOE in PTC cell lines [188]. In this study, it was found that levels of APOE expression declined with increased age in patients with PTC [188]. In addition to this finding, data showed that decreased APOE expression in PTC is associated with a statically significant decreased overall survival and found no correlation with disease-free survival [188]. This finding conflicts with other papers that found a statistically significant correlation between APOE and disease-free survival in PTC patients [189,192,194]. Further investigations concluded that mRNA levels of APOE were positively associated with the TNM staging of PTC [189,194]. Again, this was conflicting with another paper, which concluded that a decreased expression of APOE was associated with a higher TNM stage [194]. When evaluating the effects of different APOE single-nucleotide polymorphisms (SNPs), it was found that the APOE-rs429358 SNP had a positive correlation with an increased risk of PTC, whereas the SNP APOE-rs7412 had a negative correlation [191]. When stratified for age, APOE-rs429358 had a significant association in females only, while APOE-rs7412 was associated with both males and females [191]. This difference in findings suggests that, in addition to APOE’s role in ferroptosis and tumor modulation, the function of APOE-rs429358 specifically includes modulating hormonal balance in female PTC patients. However, it could also be due to the small sample size of male participants in this trial and as such, further studies are needed to validate these results [191]. The previous findings indicate that the risk of thyroid cancer and its pathogenesis can vary depending on gene polymorphisms of associated proteomic biomarkers.

An analysis of the APOE gene indicated that APOE is mostly involved in regulating cholesterol metabolism and the PPAR signaling pathway [192,195]. However, the role of APOE in PTC primarily revolves around modulating the inflammatory response in this subset of cancer patients [188,189,191]. This finding was confirmed in a further analysis that established a positive correlation between APOE expression and B cells, cytotoxic T lymphocytes, neutrophils, and dendritic cells [188,189]. Likewise, APOE is said to play a role in the activation of several cell pathways associated with cancer progression, including ferroptosis, apoptosis, DNA damage response, and intracellular signaling pathways such as PI3K/AKT, RTK, and TSC/mTOR [188,191]. APOE is a ferroptosis-related gene [191]. Ferroptosis is a recently discovered type of programmed cell death that is notable for its possible effect on the inflammatory response and its role in tumor suppression [191,196]. As such, APOE is a possible immunotherapy target in PTC patients due to the positive association with immune cell infiltration in PTC. Considering its significant elevation in PTC, APOE could be a prospective biomarker in the diagnosis of PTC [188,189]. 

Figure 4 illustrates the role of glycolysis in PTC progression [193]. It show that tumorigenesis of PTC is modulated through several mechanisms, including through the enzyme alpha-ketoglutarate-dependent dioxygenase (FTO) and its target gene, APOE [193,197]. FTO expression inhibits glycolysis in PTC and facilitates N6-methyladenosine (m6A) alteration, which is a common nucleic acid modification that ultimately affects cellular functions [197]. The m6A modification is associated with regulating tumor formation and proliferation [193]. It was established that FTO expression is reduced in PTC [193]. Reduced levels of FTO resulted in an elevation in APOE mRNA m6A alteration, which increased APOE mRNA stability [193]. This led to an increase in APOE expression. APOE promotes glycolysis in PTC through the IL-6/JAK2/STAT3 downstream signaling pathway [193]. As such, FTO suppresses PTC growth through its downstream gene target, APOE [193]. 

Similarly, cell lines from MTC were investigated to identify possible proteomic changes [181]. Using matrix-assisted laser desorption/ionization mass spectrometry imaging (MALDI-MSI), it was found that in MTC, APOE was expressed within the tumor’s amyloid components. This finding suggests that APOE could also pose as a new biomarker for the diagnosis of MTC [181]. 

Despite the previous findings, Ma et al. found that APOE did not correlate with tumor size in male PTC patients [180]. The reason for this contradiction is uncertain and requires further studies to validate these findings. Ma et al. also found that there was no association between the rate of lymph node metastasis in PTC patients and serum lipid biomarkers, including APOE [180]. Similarly, Ito et al. observed that APOE was downregulated in papillary and follicular thyroid carcinomas, whereas it was significantly overexpressed in anaplastic thyroid carcinoma based on immunohistochemical staining [195]. This finding proposes that APOE is an independent biomarker of anaplastic thyroid carcinoma. However, due to the conflicting, more recent findings, the previous conclusion requires further analysis to confirm it. 

The APOE gene is co-expressed with the APOC1 and APOC2 genes, and is also a transcriptional target gene of LXRβ [192]. As such, similar findings were discovered with regard to LXRβ upregulation in thyroid cancer, which consequently leads to an increase in APOE expression [186]. 

In patients with a diagnosis of DTC and received treatment, it was found that APOE levels remained high post-thyroidectomy and radioactive iodine (RAI) treatment, in addition to the hypothyroid state following RAI therapy [182]. These increased APOE levels remained high even following levothyroxine treatment [182]. This confirms that changes in thyroid hormone levels correlate with modifications in APO levels. 

APOL1 remains understudied, with its physiological role remaining unclear [67]. It is suggested that APOL1 mediates its effects through apoptosis and autophagy [67]. However, it was reported that APOL1 is upregulated in PTC and ATC cell lines, despite the remaining members in the APOL family remaining unchanged [67]. Table 10 presents a concise overview of the functions, regulatory mechanisms, clinical impact, prognostic importance, and origins of different apolipoproteins in thyroid cancer.

## 14. Inhibitors and Mimetic Peptides of Apolipoprotein

As outlined in the preceding sections, it is clear that apolipoproteins play diverse roles in the advancement of cancer. According to previous research and data, apolipoproteins and apolipoprotein mimetic peptides are useful as potential therapeutics due to their functions, particularly their anti-inflammatory and antioxidant properties [198]. Peptides are valued for their reduced toxicity profile and immunogenicity [198], which makes them suitable as potential therapeutics, as shown in Table 11.

APOA1 is known for its possible anti-tumorigenic characteristics. Meanwhile, mimetic peptides of APOA1 are known to remodel HDL, encourage the efflux of cholesterol from cells, and stimulate anti-inflammatory pathways, thus restricting tumor progression and improving survival, both in vitro and in vivo [198,199,200]. To test the effect of APOA1 mimetic peptides (D-4F, L-4F, and L-5F) in ovarian cancer, cell lines and mouse models were administered the peptides; this study reported a significant decrease in the serum lysophosphatidic acid levels and reduced ovarian cancer cell growth and proliferation [199,200,201]. In addition, an in vivo study in mice with ovarian cancer, the APOA1 peptide, L-5F, repressed tumor angiogenesis by inhibiting the vascular endothelial growth factor (VEGF) and basic fibroblast growth factor (bFGF) signaling pathways [202], indicating the use of L-5F as a candidate therapeutic strategy to reduce the size and number of tumor blood vessels. Another study in ovarian cancer analyzed the effect of the APOA1 peptides, L-4F and L-5F, and reported that, while L-4F repressed hypoxia-inducible factor-1α (HIF-1α) gene expression, L-5F suppressed intracellular levels of HIF-1α [203], indicating the role of the peptides in inhibiting angiogenesis and tumor growth.

Furthermore, L-4F can also suppress the tumorgenicity of cells and inflammation by reducing inflammatory cells like interleukins and ROS [204,205], supporting L-4F’s role in preventing cancer proliferation. On the other hand, research has been carried out to study the mechanism of action of the APOA1 mimetic peptide D-4F in inhibiting tumor progression [206]. D-4F was found to eliminate oxidized lipids and limit inflammatory responses [206,207] in addition to upregulating manganese superoxide dismutase (MnSOD), thereby inhibiting cancer proliferation [206,208]. These studies indicate a protective role of D-4F against lipid oxidation, and it can be considered a plausible therapeutic agent for inhibiting tumor growth and proliferation [36]. Additionally, another study used a different APOA1 mimetic peptide found in transgenic tomatoes called 6F (Tg6F). When given to mice, this study found changes in certain oxidized phospholipids, which in turn affected the expression of specific proteins like Notch and osteopontin (Spp1). These changes led to a decrease in a specific type of immune cell, called monocytic myeloid-derived suppressor cells (MDSCs), in the jejunum and lungs of the mice. These alterations were found to reduce the tumor burden in the lung, suggesting the use of oral APOA1 mimetic peptides as therapeutic agents in the intestine–lung axis [209]. 

Similarly, Mipomersen, an FDA-approved orphan drug for homozygous familial hypercholesterolemia, was developed as an antisense oligonucleotide inhibitor of APOB-100 synthesis to target and complement a specific mRNA sequence involved in coding APOB-100 [210]. The administration of Mipomersen triggers the activation of RNAse H and inhibits the microsomal triglyceride transfer protein, thereby decreasing the levels of newly synthesized APOB [210]. 

Both elevated and deficient levels of APOC2 are associated with several diseases, making it an excellent target for drug development. One study has shown that using a dipeptidyl peptidase-4 inhibitor for eight weeks significantly decreased APOC2 levels [211]. In addition, Anagliptin, a drug used for diabetes, reduced APOC2 mRNA expression in mice [212]. As for the APOC-2 mimetic peptide, a first-generation peptide, 18A-CII, was proven to regain lipolysis to normal levels in APOC2-deficient patients [213]. Due to the immunogenicity of the first generation, a second-generation mimetic peptide, D6PV, was developed; D6PV has shown a marked decline in TG in mice [214]. Another mimetic peptide, C-II-a, indicated reduced TG levels in APOC2-deficient mice [215].

In addition to the commonly studied APOA1 mimetic peptides, other apolipoprotein mimetic peptides, such as APOE mimetic peptides COG112 and OP449, were investigated [216,217]. COG112 and OP449 affected tumorigenesis in cancer cells by reducing cell viability due to their anti-inflammatory functions [216,217]. It also hinders signaling for pathogen recognition receptors (PRRs), which regulate the immune system and are implicated in cancer, in addition to reducing cell cycle progression [216]. Bhattacharjee et al. (2011) studied the anti-cancer role of APOEdp, a dimer peptide derived from the receptor-binding region of human APOE in in vitro (HUVEC cells) and in vivo models (mouse and rabbit); the study showed APOEdp to inhibit tumor growth in HUVEC cells and mice as well as restricted ocular angiogenesis [218], suggesting an anti-cancer role of APOEdp. However, APOJ mimetics suggest that they also effectively inhibit tumorigenesis [198]. Despite limited research, studies have shown that mimetic peptides of APOJ can lower lipids that promote tumor growth, thereby potentially slowing down cancer development and progression [198].

## 15. Future Research Directions

The role of apolipoproteins in the context of cancer has recently emerged as a subject of increasing interest within the scientific community. Nonetheless, it is imperative to acknowledge that notable gaps persist within the existing literature concerning the precise roles and impacts of apolipoproteins in different cancers. The available data often exhibit discrepancies, particularly with regard to the upregulation or downregulation of specific apolipoproteins, thus warranting further investigation and clarification. A substantial portion of the research dedicated to exploring the involvement of apolipoproteins in cancer has been conducted, utilizing murine models or non-human cell lines. Moving forward, a more informative and clinically relevant approach would entail a shift towards investigating the role of apolipoproteins in human cell lines.

The utilization of apolipoproteins as diagnostic tools has been well established in neurovascular and cardiovascular diseases. As demonstrated through the findings presented in this paper, apolipoproteins exhibit substantial promise as potential biomarkers for both early cancer detection and cancer prognosis in the foreseeable future. We propose that further dedicated research in this domain, with a particular focus on the prospective development of a biomarker panel consisting of apolipoproteins, would be highly beneficial. Such a panel could serve as an effective screening method for the early detection of silent cancers.

In recent times, novel applications of apolipoproteins are emerging within the scientific literature. One noteworthy path involves their potential use as therapeutic agents, as previously discussed. This can be achieved by utilizing apolipoprotein mimetic peptides or through the application of recently developed nanoparticle technologies or alternative methods. Given the considerable treatment challenges posed by numerous forms of cancer, the encouraging outcomes associated with apolipoproteins warrant further in-depth exploration. This exploration offers the prospect of uncovering novel treatment modalities or pharmaceutical interventions for cancer.

## 16. Conclusions

This review provides a comprehensive understanding of the multifaceted roles of APOs in the most prevalent cancers worldwide. Their inconsistent behavior, either increasing or decreasing in different tumor tissues, hints at a complex and not fully understood relationship with cancer. While the roles of specific APOs like APOC2, APOD, and APOM are still under investigation, continued research promises to deepen our understanding of how the APO family interacts with cancer. This could ultimately lead to the identification of new targets for cancer therapy.

## Figures and Tables

**Figure 1 cancers-15-05565-f001:**
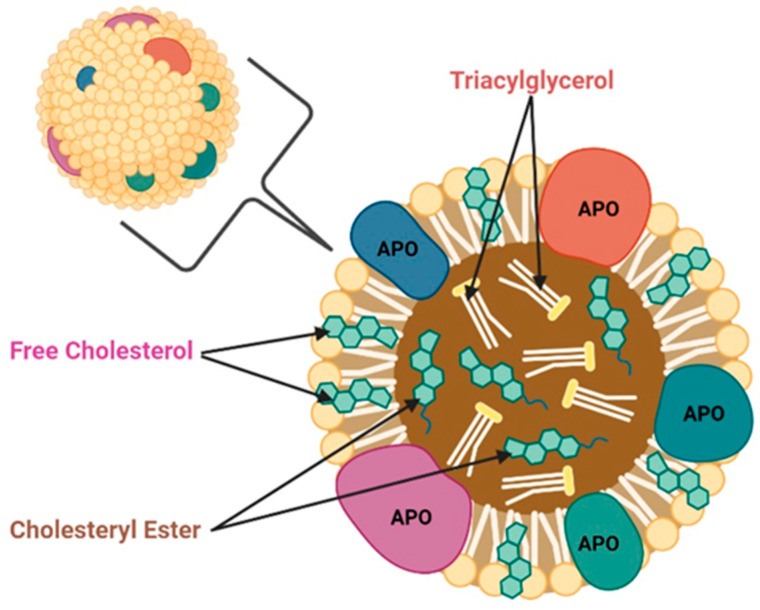
A graphical representation of an apolipoprotein illustrates its unique structure, which confers an amphipathic property to the molecule. This enables it to interact with both the lipids in the core of lipoproteins and the watery plasma environment. As a result, apolipoproteins act as biochemical keys, granting lipoprotein particles access to specific locations for the transportation, reception, or modification of lipids. Additionally, apolipoproteins play a role in stabilizing the structure of lipoproteins.

**Figure 2 cancers-15-05565-f002:**
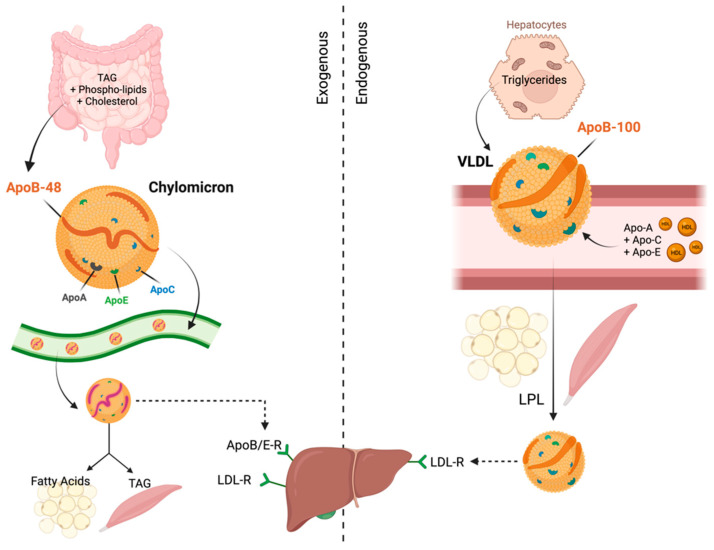
The exogenous pathway is a process for delivering triglycerides to peripheral tissues using chylomicrons and VLDLs. Chylomicron particles, consisting of APOB-48 and various other proteins, are formed by enterocytes, released into the lymph, and eventually enter the bloodstream. In peripheral tissues, chylomicron triglycerides are broken down, and the remnants are taken up by the liver, while APOA and APOC return to HDL. In the endogenous pathway, the liver produces triglycerides carried to peripheral tissues by VLDL. These VLDL particles initially contain APOB100 and acquire additional proteins and cholesteryl esters from HDL. In peripheral tissues, VLDL triglycerides are partially broken down into VLDL remnants, which are either absorbed by the liver or converted into LDL.

**Figure 3 cancers-15-05565-f003:**
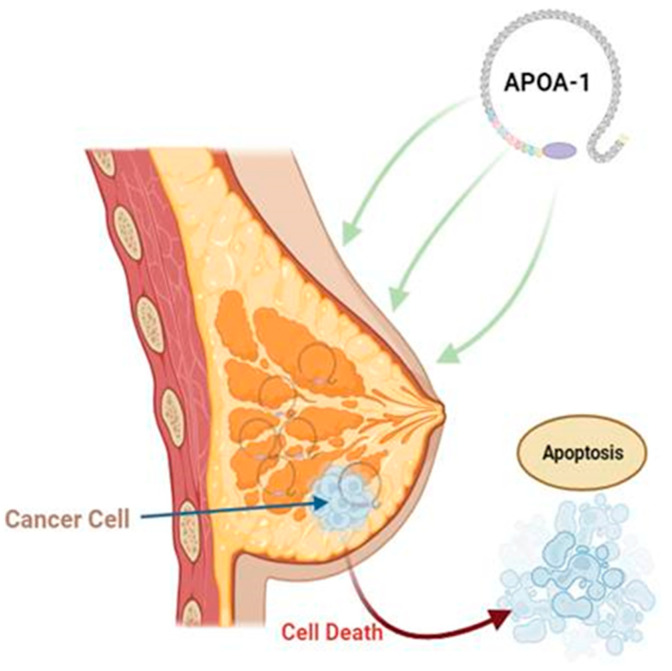
APOA-1 exerts a tumor-suppressive function in breast cancer by promoting apoptosis, thereby impeding the advancement of cancerous cells.

**Figure 4 cancers-15-05565-f004:**
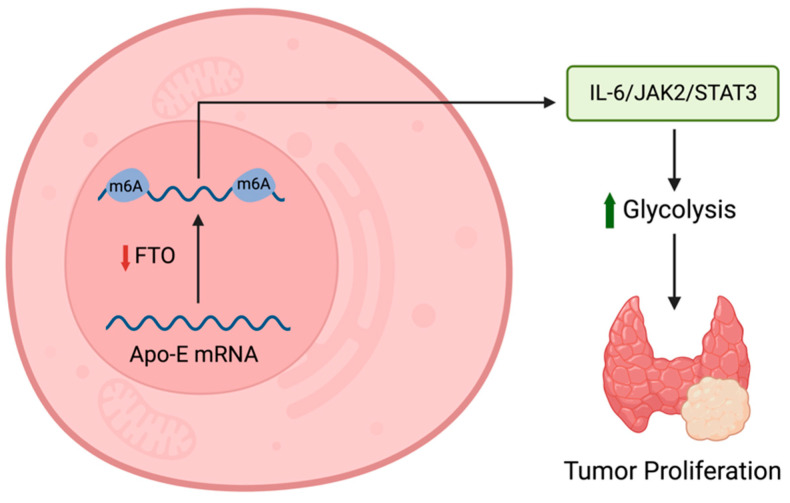
In PTC, glycolysis is crucial for tumor growth. The enzyme FTO regulates PTC growth by affecting the stability of the APOE gene [193,197]. Reduced FTO levels lead to increased APOE expression, which promotes glycolysis through the IL-6/JAK2/STAT3 signaling pathway. Thus, FTO acts as a suppressor of PTC growth [193].

**Table 1 cancers-15-05565-t001:** This table illustrates the expression of apolipoproteins (APOs) in different human cancers. ↑ indicates an increase in APO expression, while ↓ indicates a decreased expression.

Type of Cancer	APOA			APOB	APOC			APOD	APOE		APOH	APOL		APOM	APOJ
	A1	A2	A4		C1	C2	C3		E2	E4		L1	L3		
Cervical cancer	↓88							↓90							
Breast cancer	↓32			↑34	↓39	↓39	↑39	↓45			↑59			↓73	↑72
Lung cancer	↓13	↑26	↓29												
Pancreatic cancer	↓14	↓21			↑44				↑55			↑62			↑78
Nasopharyngeal cancer	↓15														
Hepatocellular cancer	↓15	↑24	↑28	↓35			↑48								↑74
Colorectal cancer	↓15			↓35	↑43			↓52			↑60		↓66	↓69	↑75
Esophageal carcinoma	↓16	↑25													
Renal cancer	↓16	↓22			↑41						↑58				↑76
Gastric cancer	↓17		↓31	↓32	↑42	↑45									↑77
Bladder cancer	↑18	↑23	↑27	↑37			↑47					↑63			
Thyroid cancer	↓19														
Ovarian cancer	↓74	↓74	↓74	↑77			↑77								↑83
Cholangiocarcinoma		↓24													
Squamous lung cancer			↑29				↑13				↑61				
Papillary thyroid cancer			↓30									↑64			
Duodenal adenoma carcinoma				↓34											
Lung NSCLC					↓40		↓7							↑67	
Prostatic cancer								↑51		↑56			↑65		↑73
Acute lymphoblastic leukemia								↓53							
Melanoma									↓54						
Acute myeloid leukemia											↑59				
Larynx cancer														↑70	

**Table 2 cancers-15-05565-t002:** Apolipoproteins in breast cancer.

Apolipoproteins in Breast Cancer						
	Level	Effect	Mechanism	Clinical Implications	Prognostic Value	Source
APOA1 [31,32]	Downregulated	Cancer cell apoptosis.	Binding of C1QBP to APOA, inhibiting its expression and APOA’s antioxidation ability.	Potential anti-tumor therapeutic target.	Reduced expression led to poor prognosis.	Unclear
APOB [34]	Upregulated	Increases metastasis.	Mutation of tumor suppressor and proto-oncogenes.	Potential prognostic marker.	Increased metastasis.	Unclear
APOC1 [39,40]	Downregulated	Anti-tumor	Inhibit cell proliferation (in vitro) and inhibit tumor growth (in vivo).	Potential diagnostic tool for differentiating between triple-negative and non-triple-negative breast cancer, as well as for early detection.	When levels decrease, there is an associated increase in tumor growth.	Hepatic
APOD [42,45,46]	Downregulated	Anti-tumor	Reduced suppression of MAPK pathway.	Potential therapeutic target.	Lower levels are associated with a higher risk of metastasis.	Localized
APOE [47,48,49,50,51,52,53,54]	Controversial	Inhibit proliferation.	High-affinity interaction with heparin and proteoglycan.	Potential prognostic biomarker.	Linked to metastasis.	Unclear
APOH [59]	Upregulated	Inhibiting apoptosis.	Unclear	Potential prognostic marker.	Linked to metastasis.	Unclear
APOJ [62]	Upregulated	Inhibiting apoptosis.	Initiate tumorigenesis	Potential prognostic marker and therapeutic target.	Linked to metastasis.	Unclear
APOL3 [70,71]	Unclear	Unclear	Regulates NCS-1, which is responsible for initiating cell metastasis and survival.	Potential prognostic marker.	Needs further research.	Unclear
APOM [72,73]	Downregulated	Reduces metastasis.	Stabilize sphingosine-1-phosphate.	Potential prognostic marker and therapeutic target.	Protective	Liver

**Table 3 cancers-15-05565-t003:** (a) Apolipoproteins in ovarian cancer. (b) Apolipoproteins in cervical cancer.

**(a) Apolipoproteins in ovarian cancer**						
	**Level**	**Effect**	**Mechanism**	**Clinical Implications**	**Prognostic Value**	**Source**
APOA1 [74,75]	Downregulated	Anti-tumor	Alters macrophage polarization towards anti-tumor M1 phenotype and inhibits angiogenesis via downregulation of MMP-9 expression.	Potential therapeutic target when induced and potential biomarker.	Associated with improved prognosis owing to protective properties.	Intestines
APOB [77,78]	Upregulated	Promotes tumor growth.	Unknown	Potential prognostic and diagnostic marker.	Linked to higher tumor grades.	Unclear
APOC3 [78]	Upregulated	Tumor metastasis.	Inhibit lipoprotein lipase and hepatic lipase.	Potential diagnostic biomarker.	Promotes malignancy.	Liver and small intestine.
APOD [80]	No significant correlation.	Improved survival.	Inhibition of tumor growth.	Potential therapeutic and prognostic target.	Linked to better prognosis when tumor is larger than 1 cm.	Unclear
APOE [81,82,83]	Upregulated	Vital for tumor growth.	Prevent cell arrest in G2 phase and apoptosis.	Potential prognostic biomarker.	Better survival when expressed.	Unclear
APOJ [84]	Upregulated	Unclear	Unknown	Diagnostic and predictive.	Linked to poor prognosis.	Unclear
**(b) Apolipoproteins in cervical cancer**						
	**Level**	**Effect**	**Mechanism**	**Clinical Implications**	**Prognostic Value**	**Source**
APOA1 [87]	Downregulated	Unclear	Anti-inflammatory, antioxidant, and anti-apoptotic.	Potential biomarker.	Decreased levels are associated with advancing stages of cancer.	Liver and small intestine.
APOC2 [88]	Downregulated in cervical cancer that caused death.	Promotes tumor growth.	Dysregulation of lipoprotein lipase.	Potential prognostic factor.	Lower levels had poorer prognosis with treatment.	Liver and intestine.
APOD [89]	Downregulated	Protective	Inhibit osteopontin-induced malignancy.	Potential therapeutic and diagnostic target.	Lower levels linked to poor prognosis.	Unclear

**Table 4 cancers-15-05565-t004:** Apolipoproteins in lung cancer.

Apolipoproteins in Lung Cancer						
	Level	Effect	Mechanism	Clinical Implications	Prognostic Value	Source
APOA1 [15,16,75,93,94,96,97,98,100]	Controversial	Anti-tumorigenic roles.	Convert cancer-linked macrophages to anti-tumor M1 phenotype, inhibit neo-angiogenesis.	Potential therapeutic target.	Linked to better prognosis.	Hepatic
APOA2 [15,16,101]	Unclear	Inflammatory marker.	Unclear	Potential diagnostic marker for early-stage lung cancer.	Requires further research.	Hepatic
APOA4 [102,103]	Variable. Upregulated in squamous cell carcinomas. Downregulated in adenocarcinomas.	Unclear	Unclear	Potential diagnostic marker.	Requires further research.	Intestinal, Tumor
APOB [93,94,104]	Variable	Controversial	Regulates cholesterol transport and metabolism.	Potential diagnostic or prognostic marker.	Controversial, may vary with tumor type.	Hepatic
APOC3 [100]	Downregulated	Unclear	Unclear	Potential diagnostic or predictive marker.	Increased levels linked to recurrence.	Hepatic
APOE [105,106,107,108,109]	Upregulated	Supports the proliferation and metastasis of lung cancer cells.	Associated with increased oxidative stress.	Potential diagnostic marker.	Linked to increased complications.	Hepatic, Unclear
APOH [103,110,111]	Upregulated	Inhibits angiogenesis.	Suppression of endothelial cell growth.	Potential therapeutic target.	Requires further research.	Unclear
APOM [72,103,112]	Downregulated in AAH. Upregulated in NSCLC.	Increased apoptosis and tumor suppression. Cell proliferation, invasion, and tumor development.	Carrier for sphingosine-1-phosphate, which inhibits ceramide, leading to cell proliferation. Inducing sphingosine-1-phosphate, which activates the ERK1/2 and PI3K/AKT signaling pathways.	Potential therapeutic target.	Requires further research.	Hepatic, Tumor
APOL2 [114]	Upregulated	Unclear	Possibly through anti-apoptotic properties.	Potential diagnostic or prognostic marker and therapeutic target.	Requires further research.	Unclear

**Table 5 cancers-15-05565-t005:** Apolipoproteins in colorectal cancer.

Apolipoproteins in Colorectal Cancer						
	Level	Effect	Mechanism	Clinical Implications	Prognostic Value	Source
APOA [7,15,30,115,116]	Upregulated	Possible anti-tumor impact.	Increases cholesterol efflux, damaging invasion distribution.	Potential therapeutic target.	Favorable factor in metastatic colorectal cancer.	Hepatic Synthesis
APOB [30,31,35,116]	Upregulated in primary CRC and with liver metastasis.	Controversial role.	May act as a lipid carrier, inactivating mutations in APOB gene, or other factors may affect its expression.	Potential therapeutic target.	Controversial, may vary with tumor stage and location.	Systemic and Hepatic
APOC1 [43,116,117]	Upregulated	Promotes cell proliferation and migration.	P38-MAPK signaling pathway activation.	Potential diagnostic or prognostic marker.	Requires further research.	Cancer Cells
APOD [7,116,118]	Downregulated in advanced stages.	Inversely connected with tumor advancement.	Reacts to oxidative stress, enhances tumor suppression through apoptosis.	Early diagnostic marker.	Linked to worse prognosis in advanced stages.	Systemic and Cancer Cells
APOE [31,116,118]	Upregulated in CRC.	Controversial role, both as a protective factor and promoter of cancer growth.	May affect intracellular adhesion and junctions, PI3K/Akt/mTOR pathway.	Potential therapeutic target.	Controversial, may vary with tumor stage and location.	Unknown
APOH [119]	Upregulated	Associated with worse prognosis.	Underlying mechanisms unknown.	Requires further research.	Linked to worse prognosis.	Unknown
APOJ [7,75,79]	Upregulated in colon cancer.	Promotes colorectal carcinogenesis, metastasis, and tumor invasion.	Activated by stress, pro- and anti-apoptotic activities.	Potential diagnostic or prognostic marker.	Requires further research on mechanisms.	Cancer Cells
APOM [7,69,120]	Controversial	Controversial role.	May inhibit EMT or promote cell proliferation and invasion through the RPS27A-MDM2-p53 pathway.	Potential diagnostic or prognostic marker.	Controversial, may vary with tumor stage and grade.	Systemic and Cancer Cells

**Table 6 cancers-15-05565-t006:** Apolipoproteins in pancreatic cancer (PC).

Apolipoproteins in Pancreatic Cancer						
	Level	Effect	Mechanism	Clinical Implications	Prognostic Value	Source
APOA [31,122,123,124,125,126]	Upregulated in early stages and downregulated in later stages.	Inhibits tumor progression.	Underlying mechanisms unknown.	Potential therapeutic target, and prognostic and diagnostic marker.	Low levels of APOA1 are linked to worse prognosis, while high levels of APOA4 indicate worse prognosis.	Tumor cells
APOC [7,31]	Upregulated	APOC1 induces apoptosis while APOC2 promotes cell proliferation and invasion.	Requires further research.	Potential therapeutic target and prognostic marker.	High levels of APOC1 are linked to worse survival.	Tumor cells
APOE1 [128,129]	Upregulated	Inhibits apoptosis of malignant cells.	Activation of the NF-κB signaling pathway and production of CXCL1.	Potential therapeutic target and prognostic marker.	High levels are linked to worse prognosis.	Tumor cells
APOE2 [130,131]	Upregulated	Prevents mitochondrial apoptosis and promotes tumor growth.	Increased expression of BCL-2 through activating ERK1/2/CREB signaling cascade.	Potential prognostic marker and therapeutic target for PC.	Requires further research.	Tumor cells
APOL1 [135,136]	Controversial	Complex role; can both induce and inhibit proliferation and apoptosis.	Can inhibit proliferation and induce apoptosis by activating the NOTCH1 pathway.	Potential therapeutic target.	Requires further research.	Human pancreatic ductal adenocarcinomal cell lines.
APOJ [132,134]	Further research is needed	Complex role as it regulates a diversity of pathways.	Modulates a variety of signaling pathways such as ERK, AKT, and NF-κB.	Potential target for treatment.	Requires further research.	Tumor cells

**Table 7 cancers-15-05565-t007:** Apolipoproteins in hepatic cancer.

Apolipoproteins in Hepatic Cancer						
	Level	Effect	Mechanism	Clinical Implications	Prognostic Value	Source
APOA1 [137,138,139,140]	Upregulated in early stages of hepatic cancer.	Suppresses immunity and promotes cell proliferation.	Requires further research.	Potential therapeutic target and prognostic marker.	Predictor of liver metastasis and resistance to PD-1 inhibitor treatment.	Hepatic
APOB [31,142,143]	Upregulated in hepatic cancer and liver metastasis.	Requires further research.	Unknown	Potential therapeutic target and prognostic marker.	Predictor of liver metastasis and resistance to PD-1 inhibitor treatment.	Hepatic
APOC [144,145,146,147]	Upregulated in hepatic cancer and liver metastasis.	APOC1 can promote immune evasion and angiogenesis2 of tumor cells.	APOC1 can trigger the transformation of tumor-associated macrophages (TAMs) into M2-like cells.	Potential prognostic marker.	APOC1 and APOC4 are associated with overall survival, APOC3 is associated with both overall survival and recurrence-free survival.	Hepatic and tumor cells.
APOJ [150,151,152]	Controversial	Promotes autophagy and metastasis.	Upregulation of NF-κB and Beclin1, promotion of ER stress, and activation of PERK and ATF6 signaling pathways.	Potential therapeutic target.	Linked with resistance to sorafenib/doxorubicin.	Hepatic

**Table 8 cancers-15-05565-t008:** Apolipoproteins in prostate cancer.

Apolipoproteins in Prostate Cancer						
	Level	Effect	Mechanism	Clinical Implications	Prognostic Value	Source
APOA [153]	Upregulated, increases as disease progresses.	Enhances cell proliferation, invasiveness, and resistance to therapeutics.	MYC regulates APOA expression.	Potential prognostic marker.	High levels of APOA1 indicate disease progression.	Tumor cells
APOC [155,156]	Upregulated	APOC1 inhibits apoptosis and promotes proliferation.	APOC1 promotes survivin/phosphor-Rb/p21 pathway, reducing caspase-3, inhibiting apoptosis.	Potential therapeutic target and prognostic marker.	Elevated APOC1 levels are linked to worse prognosis.	Tumor cells
APOD [157]	Upregulated	Requires further research.	Requires further research.	Potential cellular marker.		Tumor cells
APOE [158]	Upregulated	Inhibits immune reaction against tumor cells.	Binds TREM2 on neutrophils, inducing senescence.	Potential therapeutic target and prognostic marker.	Elevated expression is linked to poor prognosis.	Tumor cells
APOL3 [69]		Increases hereditary prostate cancer susceptibility.	Suggests the role of 22q locus in sporadic prostate cancer.	Potential risk estimator.	Requires further research.	Patient samples
APOJ [159]	Upregulated	Inhibits apoptosis. Helps tumor cells evade the effects of androgen ablation and chemotherapeutic agents.	Requires further research.	Potential target to increase chemosensitivity.	Requires further research.	Tumor cells

**Table 9 cancers-15-05565-t009:** Apolipoproteins in gastric cancer.

Apolipoproteins in Gastric Cancer						
	Level	Effect	Mechanism	Clinical Implications	Prognostic Value	Source
APOA1 [160,161]	Controversial	Increases tumor burden.	Requires further research.	After gastrectomy, APOA-1 significantly decreases. May be used to differentiate between chronic gastritis and gastric cancer.	Circulating APOA-1 levels reflect tumor burden.	Tumor cells
APOA2 [162]	Upregulated in high-claudin-6 gastric cancers.	Affects cholesterol metabolism.	Requires further research.	Effective prognostics marker.		Tumor cells
APOC1-3 [163,164,165,166,167,168]	Controversial	Inhibit apoptosis and promotes proliferation.	CD36-mediated PI3K/AKT/mTOR signaling pathway.	Potential therapeutic target and prognostic marker.	Elevated APOC1 levels are linked to worse prognosis. APOC2 is elevated in peritoneal metastasis.	Unknown
APOE [169,170]	Upregulated	Cancer development and progression.	Requires further research.	Elevated levels are associated with increased risk of muscular invasion.	Shorter survival time in patients with elevated APOE.	
APOJ [171]	Upregulated	Tumor progression and metastasis.	Requires further research.	Potential therapeutic target.	Requires further research.	

**Table 10 cancers-15-05565-t010:** Apolipoproteins in thyroid cancer.

Apolipoproteins in Thyroid Cancer						
	Level	Effect	Mechanism	Clinical Implications	Prognostic Value	Source
APOA [15,16,172,173,174,175,176,180]	Downregulated	Dysregulated lipid profile and altered gut microbiota symbiosis.	Regulates lipid proteomic profiles. LXR/RXR activation pathway.	Potential diagnostic or prognostic marker.	APOA1 linked to smaller tumor size. Reduced APOA1 levels linked to worse prognosis and aggressive tumor characteristics.	Hepatic, Intestinal
APOB [15,16,183]	Downregulated	Unclear	Unclear, hypothesized to be through lipid pathways.	Potential diagnostic or prognostic marker.	Reduced APOB linked with more aggressive tumors.	Hepatic
APOC [184,185,186]	Controversial	Unclear, possibly leads to tumor progression.	Orphan nuclear hormone receptor superfamily receptors bind to HRE RXRalpha and T3Rbeta LXRβ overexpression.	Potential prognostic biomarker.	APOC decreases as cancer stage increases. APOC linked to poor tumor characteristics.	Hepatic, possibly tumor.
APOD [187]	Downregulated in DTC.	Increased tumor proliferation.	P53 tumor suppressor family.	Potential prognostic marker.	Linked to higher risk score and recurrence.	Unclear
APOE [181,186,188,189,190,191,192,193]	Upregulated	Tumor progression.	Programmed cell death and tumor modulation. Modulating the inflammatory response.	Potential prognostic biomarker. Possible therapeutic target.	Linked to overall survival.	Hepatic, Tumor
APOL1 [67]	Upregulated	Unclear	Apoptosis and autophagy.	Potential diagnostic marker and therapeutic target.	Requires further research.	Unclear

**Table 11 cancers-15-05565-t011:** Therapeutic agents of APOs.

APO	Therapeutic Agent	Effect	Mechanism of Action
APO Mimetics
APOA1	D-4F	Decreases cancer proliferation.	Eliminates oxidized lipids.Limits inflammatory responses.Upregulates MnSOD [1,2].
L-4F	Represses tumor angiogenesis, tumorigenicity of cell and inflammation.	Represses HIF-1α.Reduces interleukins and ROS [3,4].
L-5F	Represses tumor angiogenesis.	Inhibits VEGF and bFGF signaling pathways [5]. Suppresses intracellular levels of HIF-1α [6].
Tg6F	Reduces MDSC in jejunum and lung.	Alters expression of Notch and Spp1 [7].
APOC2	18A-CII	Regains lipolysis to normal levels in APOC-II deficient patients [8].	
D6PV [9]	Decreases TG levels.	
C-II-a [10]	
APOE	COG112	Anti-inflammatory and reduces cell cycle progression [11,12].	Hinders signaling for PRR.
OP449
APOEdp	Inhibit tumor growth and restrict ocular angiogenesis [13]	
APOJ	G*	Decreases tumorigenesis [14].	Decreases pro-tumorigenic lipids [14].
**APO Inhibitors**
APOB-100	Mipomersen	Decreases levels of newly synthesized APOB-100.	Triggers activation of RNase H and inhibits microsomal triglyceride transfer protein [15].
APOC2	DPP4 Inhibitor [16]	Decreases APOC-II levels.	
Anagliptin [17]	Decreases APOC-II mRNA expression.

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
