# Peer review of "The Role of Apolipoproteins in the Commonest Cancers: A Review"

_cancers, 2023, doi:10.3390/cancers15235565_

Round 1

Reviewer 1 Report

Comments and Suggestions for Authors

I have had the opportunity to thoroughly review the manuscript titled "The Role of Apolipoproteins in Commonest Cancers: A Review." Overall, I appreciate the author's attempt to explore the intriguing connection between apolipoproteins and cancer, which is an area of growing interest in the field of oncology. However, there are several strengths and areas for improvement that should be considered before publication.

Strengths:

·        Comprehensive Literature Review: The manuscript exhibits a commendable effort in compiling and summarizing a wide range of studies related to apolipoproteins and various types of cancer. This comprehensive review of existing research provides a solid foundation for understanding the subject matter.

·        Clear Writing Style: The writing style is generally clear and accessible, making the manuscript suitable for a broad audience, including researchers and healthcare professionals who may not have a specialized background in lipid biology or cancer.

·        Inclusion of Relevant Studies: The manuscript appropriately references and discusses recent studies and findings, which enhances its relevance and credibility.

Areas for Improvement:

·        Structural Organization: The manuscript could benefit from improved structural organization. While the introduction is well-written and engaging, the subsequent sections lack a clear thematic progression. It would be helpful to establish a logical flow, perhaps by organizing the discussion around specific apolipoproteins and their roles in different cancer types.

·        Critical Analysis: While the manuscript provides a summary of existing research, it lacks a deeper critical analysis of the findings. For instance, it would be valuable to discuss conflicting results, limitations in study designs, and potential sources of bias in the reviewed studies. This critical evaluation would contribute to a more nuanced understanding of the topic.

·        Concise Summaries: Consider including concise summaries or tables that highlight the key findings for each cancer type discussed. This would aid readers in quickly grasping the most relevant information without needing to sift through dense paragraphs of text.

·        Future Research Directions: It would be beneficial to conclude the manuscript with a section on potential future research directions. This could include areas where further investigation is needed, emerging trends, and unanswered questions in the field.

·        Clarity of Figures and Tables: If you plan to include figures and tables in the final manuscript, ensure that they are clear, well-labeled, and directly support the text. Visual aids can greatly enhance comprehension.

·        Citation Consistency: Ensure consistency in citation style throughout the manuscript. Verify that all references follow a uniform citation format, such as APA, Chicago, or another appropriate style.

In conclusion, "The Role of Apolipoproteins in Commonest Cancers: A Review" has the potential to make a valuable contribution to the field of cancer research. To maximize its impact, I recommend addressing the structural and analytical issues mentioned above and carefully revising the manuscript accordingly. With these improvements, the manuscript will provide a more comprehensive and critical exploration of the subject, benefiting both researchers and healthcare practitioners in the field. I look forward to seeing the revised version.

Comments on the Quality of English Language

·        Citation Consistency: Ensure consistency in citation style throughout the manuscript. Verify that all references follow a uniform citation format, such as APA, Chicago, or another appropriate style.

Author Response

Dear Reviewer,

Thank you for your insightful feedback.

Please see authors point-by-point response at below:

Comments and Suggestions for Authors

I have had the opportunity to thoroughly review the manuscript titled "The Role of Apolipoproteins in Commonest Cancers: A Review." Overall, I appreciate the author's attempt to explore the intriguing connection between apolipoproteins and cancer, which is an area of growing interest in the field of oncology. However, there are several strengths and areas for improvement that should be considered before publication.

Strengths:

  • Comprehensive Literature Review: The manuscript exhibits a commendable effort in compiling and summarizing a wide range of studies related to apolipoproteins and various types of cancer. This comprehensive review of existing research provides a solid foundation for understanding the subject matter.
  • Clear Writing Style: The writing style is generally clear and accessible, making the manuscript suitable for a broad audience, including researchers and healthcare professionals who may not have a specialized background in lipid biology or cancer.
  • Inclusion of Relevant Studies: The manuscript appropriately references and discusses recent studies and findings, which enhances its relevance and credibility.

Point 1: Areas for Improvement:

  • Structural Organization: The manuscript could benefit from improved structural organization. While the introduction is well-written and engaging, the subsequent sections lack a clear thematic progression. It would be helpful to establish a logical flow, perhaps by organizing the discussion around specific apolipoproteins and their roles in different cancer types.

Response 1: Thank you for your insightful feedback regarding our manuscript's organization. We have carefully considered your suggestion to arrange the content around specific apolipoproteins and their roles across various cancers. Our intention is to present a novel perspective on apolipoproteins in cancer by focusing on the role they play within individual cancer types. This structure not only offers a deep dive into each cancer's specificities but also aids in understanding the therapeutic and diagnostic potentials of apolipoproteins in a disease-centric manner. We acknowledge this approach deviates from traditional classifications, yet we believe it enriches the content for readers with an interest in particular cancers, providing a holistic view of the apolipoproteins involved. To balance thematic clarity with our chosen structure, we have added additional tables summarizing the roles, control mechanisms, clinical impacts, prognostic relevance, and derivation of different apolipoproteins per cancer type, improving navigation and comprehension. We trust these revisions address your concerns and uphold the manuscript's distinctiveness. We remain open to further feedback to ensure the paper's clarity and utility.

Point 2:  Critical Analysis: While the manuscript provides a summary of existing research, it lacks a deeper critical analysis of the findings. For instance, it would be valuable to discuss conflicting results, limitations in study designs, and potential sources of bias in the reviewed studies. This critical evaluation would contribute to a more nuanced understanding of the topic.

Response 2: Thank you for the insightful feedback on enhancing the critical analysis in our manuscript. We have revisited and amended the manuscript to include a thorough discussion on the varying findings related to apolipoproteins' roles in different cancers. Additionally, we have identified and detailed the limitations and possible biases in the study designs of the reviewed literature to provide a clearer, more accurate interpretation of the research.

Point 3:  Concise Summaries: Consider including concise summaries or tables that highlight the key findings for each cancer type discussed. This would aid readers in quickly grasping the most relevant information without needing to sift through dense paragraphs of text.

Response 3: We are grateful for your recommendation to include brief summaries in our manuscript. In line with your advice, we have now incorporated concise tables in each section. These tables provide quick insights into the pivotal findings for each type of cancer, enabling readers to rapidly comprehend the core points of our review. We trust that these tables will make for an effective guide and improve both the accessibility and the clarity of our paper for the reader.

Point 4:  Future Research Directions: It would be beneficial to conclude the manuscript with a section on potential future research directions. This could include areas where further investigation is needed, emerging trends, and unanswered questions in the field.

Response 4: We acknowledge the importance of guiding future research and, in accordance with your suggestion, have added a new section at the end of the manuscript dedicated to future research directions. This section identifies and elaborates on gaps in the current knowledge base and proposes specific areas that would benefit from further exploration. We hope this addition will not only serve as a springboard for future studies but also stimulate continued scholarly discussion within the field (see page 31, under “15. Future Research Directions.”)

Point 5:  Clarity of Figures and Tables: If you plan to include figures and tables in the final manuscript, ensure that they are clear, well-labeled, and directly support the text. Visual aids can greatly enhance comprehension.

Response 5: In response to your feedback, we have taken meticulous care to ensure that all figures and tables included in the final manuscript are clear, well-labeled, and effectively complement the accompanying text. We have added new two new figures specifically in the sections detailing Apolipoproteins in Normal Cells (Figure 2) and Apolipoproteins in Thyroid Cancer (Figure 4) to strengthen the comprehension. These visual aids have been designed to provide at-a-glance insights into the discussed content and facilitate an easier grasp of complex information. We have also included additional Tables throught the paper to offers a brief encapsulation of the roles, regulatory frameworks, clinical outcomes, prognostic implications, and sources of various apolipoproteins in per cancer type.

Point 6:   Citation Consistency: Ensure consistency in citation style throughout the manuscript. Verify that all references follow a uniform citation format, such as APA, Chicago, or another appropriate style.

Response 6: We have thoroughly reviewed and updated the citations within our manuscript to ensure uniformity in the citation style as per the journal's guidelines. This consistency has been meticulously maintained across all referenced works.

In conclusion, "The Role of Apolipoproteins in Commonest Cancers: A Review" has the potential to make a valuable contribution to the field of cancer research. To maximize its impact, I recommend addressing the structural and analytical issues mentioned above and carefully revising the manuscript accordingly. With these improvements, the manuscript will provide a more comprehensive and critical exploration of the subject, benefiting both researchers and healthcare practitioners in the field. I look forward to seeing the revised version.

Comments on the Quality of English Language

  • Citation Consistency: Ensure consistency in citation style throughout the manuscript. Verify that all references follow a uniform citation format, such as APA, Chicago, or another appropriate style.

Reviewer 2 Report

Comments and Suggestions for Authors

This manuscript provides a comprehensive review of the various isoforms of apolipoproteins and their association with the incidence of various cancers, especially focusing on the most common types including breast, lung, gynecological, colorectal, thyroid, gastric, pancreatic, hepatic, and prostate cancers. The potential clinical relevance of various apolipoprotein inhibitors is also discussed in depth. While the  content is well-organized and categorized according to cancer types, there are several parts that could be enhanced. 

The manuscript's theme and content are very similar to that of a cited article (reference 119, line 1114, page 23) which also discusses the role of apolipoprotein isoforms in the development of cancer. I recommend that the author introduce more unique perspectives or novel discussions to distinguish this manuscript from the cited work, thereby adding originality.

For example, the article mentions that not all apolipoproteins are derived from the liver and circulate throughout the body; some may be expressed by cancer cells themselves, possibly having a localized effect on promoting or inhibiting cancer cell growth and development. The authors could check all cited literatures to ascertain the sources of apolipoproteins and classify them as systemic, hepatic, or expressed by the cancer cells themselves. 

The manuscript is heavily text-based without corresponding conceptual diagrams. Only one figure illustrating the role of apolipoproteins in apoptosis in breast cancer cells is provided. I recommended that the authors incorporate additional conceptual diagrams to cover all content discussed in the manuscript, facilitating a better understanding for the readers.

Author Response

Dear Reviewer,

Thank you for your insightful feedback.

Please see authors point-by-point response at below:

Comments and Suggestions for Authors

This manuscript provides a comprehensive review of the various isoforms of apolipoproteins and their association with the incidence of various cancers, especially focusing on the most common types including breast, lung, gynecological, colorectal, thyroid, gastric, pancreatic, hepatic, and prostate cancers. The potential clinical relevance of various apolipoprotein inhibitors is also discussed in depth. While the  content is well-organized and categorized according to cancer types, there are several parts that could be enhanced.

Point 1: The manuscript's theme and content are very similar to that of a cited article (reference 119, line 1114, page 23) which also discusses the role of apolipoprotein isoforms in the development of cancer. I recommend that the author introduce more unique perspectives or novel discussions to distinguish this manuscript from the cited work, thereby adding originality.

Response 1: We acknowledge the concerns regarding the similarities between our manuscript and the article referenced (reference 119, line 1114, page 23). To address this, we have taken significant steps to differentiate our work and add original value. We have included concise tables summarizing pivotal data for each cancer type discussed, offering a unique at-a-glance comparison not present in the referenced article. Additionally, we have appended a section forecasting potential avenues for future research, providing guidance for upcoming studies in the field. To further individualize our manuscript, we have conducted a nuanced critical analysis of existing literature, including the studies cited, to draw more specific and relevant conclusions. We believe these enhancements substantially increase the originality and scholarly contribution of our paper.

Point 2: For example, the article mentions that not all apolipoproteins are derived from the liver and circulate throughout the body; some may be expressed by cancer cells themselves, possibly having a localized effect on promoting or inhibiting cancer cell growth and development. The authors could check all cited literatures to ascertain the sources of apolipoproteins and classify them as systemic, hepatic, or expressed by the cancer cells themselves.

Response 2: We appreciate your insightful suggestion regarding the origin of apolipoproteins. In accordance with your guidance, we have meticulously revisited our cited literature to accurately classify the sources of apolipoproteins. The revised tables at the end of each cancer-specific section now include a designated column to clearly indicate the source for each apolipoprotein.This addition highlights the potential localized effects of apolipoproteins on cancer cell behavior and contributes to a more detailed understanding of their roles in cancer biology.

Point 3: The manuscript is heavily text-based without corresponding conceptual diagrams. Only one figure illustrating the role of apolipoproteins in apoptosis in breast cancer cells is provided. I recommended that the authors incorporate additional conceptual diagrams to cover all content discussed in the manuscript, facilitating a better understanding for the readers.

Response 3: In response to the concern about our manuscript's density of text, we have added two new figures—Figure 2 in the 'Apolipoproteins in Normal Cells' section and Figure 4 in the 'Apolipoproteins in Thyroid Cancer' section—to aid understanding. These figures distill the content into visual form for immediate comprehension of complex topics. Additionally, we have interspersed tables throughout the document summarizing the roles, mechanisms, clinical relevance, and sources of apolipoproteins for each cancer type examined.

Round 2

Reviewer 2 Report

Comments and Suggestions for Authors

The text and figures is now clear and considerably enhanced, meeting publishing quality.